# CD4 expression in effector T cells depends on DNA demethylation over a developmentally established stimulus-responsive element

Athmane Teghanemt[1,2], Priyanjali Pulipati [1,2,10], Kara Misel-Wuchter [1,3,10], Kenneth Day[4], Matthew S. Yorek[1,2], Ren Yi [5], Henry L. Keen[6], Christy Au[7], Thorsten Maretzky [1,2], Prajwal Gurung[1,2], Dan R. Littman[7,8] & Priya D. Issuree [1,2,3,9✉]

The epigenetic patterns that are established during early thymic development might determine mature T cell physiology and function, but the molecular basis and topography of the genetic elements involved are not fully known. Here we show, using the *Cd4* locus as a paradigm for early developmental programming, that DNA demethylation during thymic development licenses a novel stimulus-responsive element that is critical for the maintenance of *Cd4* gene expression in effector T cells. We document the importance of maintaining high CD4 expression during parasitic infection and show that by driving transcription, this stimulus-responsive element allows for the maintenance of histone H3K4me3 levels during T cell replication, which is critical for preventing de novo DNA methylation at the *Cd4* promoter. A failure to undergo epigenetic programming during development leads to gene silencing during effector T cell replication. Our study thus provides evidence of early developmental events shaping the functional fitness of mature effector T cells.

[1] Inflammation Program, Roy J. and Lucille A, Carver College of Medicine University of Iowa, Iowa City, IA 52242, USA. [2] Department of Internal Medicine, Roy J. and Lucille A, Carver College of Medicine University of Iowa, Iowa City, IA 52242, USA. [3] Molecular Medicine Graduate Program, Iowa City, IA 52242, USA. [4] Zymo Research Corporation, Irvine, CA, USA. [5] Department of Computer Science, New York University, New York, NY 10011, USA. [6] Bioinformatics Division of the Iowa Institute of Human Genetics, University of Iowa, Iowa City, IA 52242, USA. [7] The Kimmel Center for Biology and Medicine of the Skirball Institute, New York University School of Medicine, New York, NY 10016, USA. [8] Howard Hughes Medical Institute, New York, NY 10016, USA. [9] Immunology Graduate Program, Iowa City, IA 52242, USA. [10] These authors contributed equally: Priyanjali Pulipati, Kara Misel-Wuchter. ✉email: Priya-issuree@uiowa.edu

CD4[+] T cells undergo dynamic and extensive changes in their transcriptional and epigenetic landscapes during cell fate specification from bipotent CD4[+]CD8[+] double-positive (DP) precursors in the thymus[1]. One of the critical epigenetic marks that is dynamically modulated in CD4[+] T cells is DNA methylation, as illustrated by the abundant presence of the first intermediate of active DNA demethylation, 5hmC, in newly specified CD4[+]CD8[−] thymocytes[2]. A large proportion of these 5hmC marks occur predominantly in intergenic and intragenic regions of active genes but are depleted at the transcription start sites (TSS)[2]. However, the roles of 5mC/5hmC at these nonpromoter and noncoding sites remain to be elucidated. As DNA methylation is a stable and heritable mark that can be passed on to daughter cells, an important question that arises is whether early thymic developmental changes play a role in the programming of gene expression and phenotypes later during effector helper T-cell differentiation.

To discern the long-lasting effects of early epigenetic programming, we took advantage of the *Cd4* gene that undergoes extensive epigenetic programming during thymic differentiation[3,4]. In addition to being a critical regulator of T-cell receptor signaling and helper-lineage specification[5,6], CD4 also serves as a tractable marker linked to effector helper T-cell programs. Previously, we showed that as DP precursors commit to the CD4[+]CD8[−] lineages, the *Cd4* locus undergoes extensive TET1/3-mediated DNA demethylation and this demethylation process is highly coordinated via stage-specific activities of *cis*-regulatory elements (CRE) at the locus[5]. An upstream CRE (E4p), which dictates *Cd4* expression in DP precursors, and a downstream intronic CRE (E4m), which modulates *Cd4* expression during commitment and maturation of helper and regulatory T cells, are both required for TET1/3-mediated demethylation[5,7,8]. A failure to undergo demethylation during development significantly reduces CD4 expression in dividing effector T cells, but does not compromise CD4 expression in naive and mature CD4[+] T cells[5]. The mechanism of why expression is only dramatically affected in effector T cells remains unclear.

Herein, we report that early epigenetic programming during T-cell development plays a critical role in the optimal licensing of stimulus-responsive elements that maintain gene-expression programs in effector CD4[+] T cells. We identify a novel stimulus-responsive CRE (E4a) that modulates *Cd4* expression in a partially redundant manner in effector helper T cells. We show that E4a is licensed early during thymic T-cell development for its function later in effector T cells through intergenic and intragenic DNA demethylation. A lack of DNA demethylation during thymic development significantly compromises its function in effector cells and leads to an inability to maintain H3K4me3 levels at the *Cd4* promoter during cellular replication. This results in aberrant de novo DNA methylation at the promoter, leading to gene silencing, and impairs Th1 differentiation and parasitic clearance in a mouse model of cutaneous Leishmaniasis. Furthermore, using a genome-wide analysis, we find that epigenetic programming during T-cell development modulates maintenance of gene expression of a broad number of genes in effector CD4[+] T cells. Taken together, our study provides a mechanistic demonstration of the potential for early epigenetically transmitted transcriptional programs in modulating post-developmental T cell physiology, which may have important implications for the ontogeny of autoimmune disease.

## Results

### E4a modulates *Cd4* expression in effector T cells in a partially redundant manner with E4m.

We and others previously demonstrated a role for the E4m CRE in the upregulation of CD4 expression following positive selection in the thymus[5,9]. Together with another CRE, E4p, which controls CD4 expression in DP thymocytes[7], E4m ensures sufficient CD4 expression for continuous and robust TCR signaling during lineage commitment, and is vital to ensure differentiation into helper and regulatory T-cell lineages[5,9]. E4m-deficient T cells in the periphery continued to display reduced CD4 expression, indicating that this enhancer function may be required to set the stage for optimal expression post-thymic maturation. In addition, CD4 expression in *Cd4*[E4mΔ/Δ] T cells was further reduced in the course of T-cell proliferation[5], indicating a possible role for E4m-dependent epigenetic regulation of *Cd4*. To first test the requirement of E4m in post-thymic T cells, we generated *Cd4*[E4mflox/flox] mice. Using a bicistronic Cre-recombinase GFP retroviral-transduction approach[7], we induced recombination of E4m in in vitro activated CD4[+] T cells. Unexpectedly, CD4 expression was not significantly affected in proliferating cells following deletion of E4m (Fig. 1a, b, Supplementary Fig. 1a) suggesting that E4m is dispensable for maintaining CD4 expression post-thymically. We hypothesized that another CRE may compensate for maintaining CD4 expression in proliferating T cells in the absence of E4m. To test this, we used the Immgen ATAC-seq database to examine chromatin accessibility, as a readout for putative CREs in CD4[+] thymic T cells, CD4[+] naive T cells, and in vitro-activated CD4[+] T cells, and identified a region upstream of the *Cd4* TSS that was uniquely accessible in activated CD4[+] T cells (Fig. 1c). We designated this putative CRE as E4a and generated *Cd4*[E4aΔ/Δ] and *Cd4*[E4aΔ/Δ/ E4mΔ/Δ] mice. CD4[+] T cells from *Cd4*[E4aΔ/Δ] mice displayed a modest but significant reduction in CD4 expression upon activation (Supplementary Fig. 1b, c). However, combined loss of E4m and E4a resulted in an earlier and more profound loss of CD4 expression than loss of E4m alone (Fig. 1d, e). Strikingly, CD4 expression in *Cd4*[E4aΔ/Δ/ E4mΔ/Δ] T cells was markedly reduced following proliferation at 48 h as compared with 16 h post T-cell activation, during which time there is no cell division (Supplementary Fig. 1d), suggesting a replication-coupled loss of expression. The profound reduction in CD4 expression in *Cd4*[E4aΔ/Δ/ E4mΔ/Δ] T cells was accompanied by reduced *Cd4* mRNA expression, consistent with compromised transcription (Fig. 1f). We further assessed whether loss of CD4 was due to alternately spliced *Cd4* transcripts, but found similar defects in mRNA levels when assessing usage of exons 1, 2, and 3 (Supplementary Fig. 1e, f). Loss of CD4 expression in *Cd4*[E4mΔ/Δ/ E4aΔ/Δ] T cells was independent of the strength of TCR activation, as serial dilution of anti-CD3 stimulation led to equal loss of CD4 expression during proliferation (Supplementary Fig. 1g, h). Furthermore, when naive CD4 helper T cells from *Cd4*[E4mΔ/Δ/ E4aΔ/Δ] were cotransferred with WT naive CD45.1 helper T cells into *Rag*[−/−] mice (Fig. 1g), CD45.1 WT T cells maintained high CD4 levels during homeostatic proliferation, while *Cd4*[E4aΔ/Δ/ E4mΔ/Δ] T cells showed a dramatic downregulation (Fig. 1h, i). Together, our data suggest that E4a is a stimulus-responsive CRE and acts in a partially redundant manner with E4m to maintain CD4 expression in dividing CD4[+] T cells.

### E4a is a stimulus-responsive *cis*-regulatory element licensed during development.

We next assessed whether E4a was required during T cell development. Deletion of E4a alone had no effect on CD4 expression in preselected TCRβ[lo]CD24[hi]CD69[−] DP thymocytes, nor in recently selected CD4[+]TCRβ[hi]CD24[hi]CD69[+], mature CD4[+]TCRβ[hi]CD24[lo]CD69[−] thymocytes or naive peripheral CD4[+] T cells (Supplementary Fig. 2a, b). *Cd4*[E4mΔ/Δ/ E4aΔ/Δ] mice mirrored *Cd4*[E4mΔ/Δ] mice with normal CD4 levels in preselected DP thymocytes and reduced expression in recently selected and mature CD4[+] thymocytes compared with *Cd4*[+/+] controls (Fig. 2a, b). In peripheral CD4 T cells, *Cd4*[E4mΔ/Δ/ E4aΔ/Δ] mice displayed a modest but statistically significant reduction in CD4 expression compared with *Cd4*[E4mΔ/Δ] mice (Fig. 2c, d, Supplementary Fig. 2c, d). These

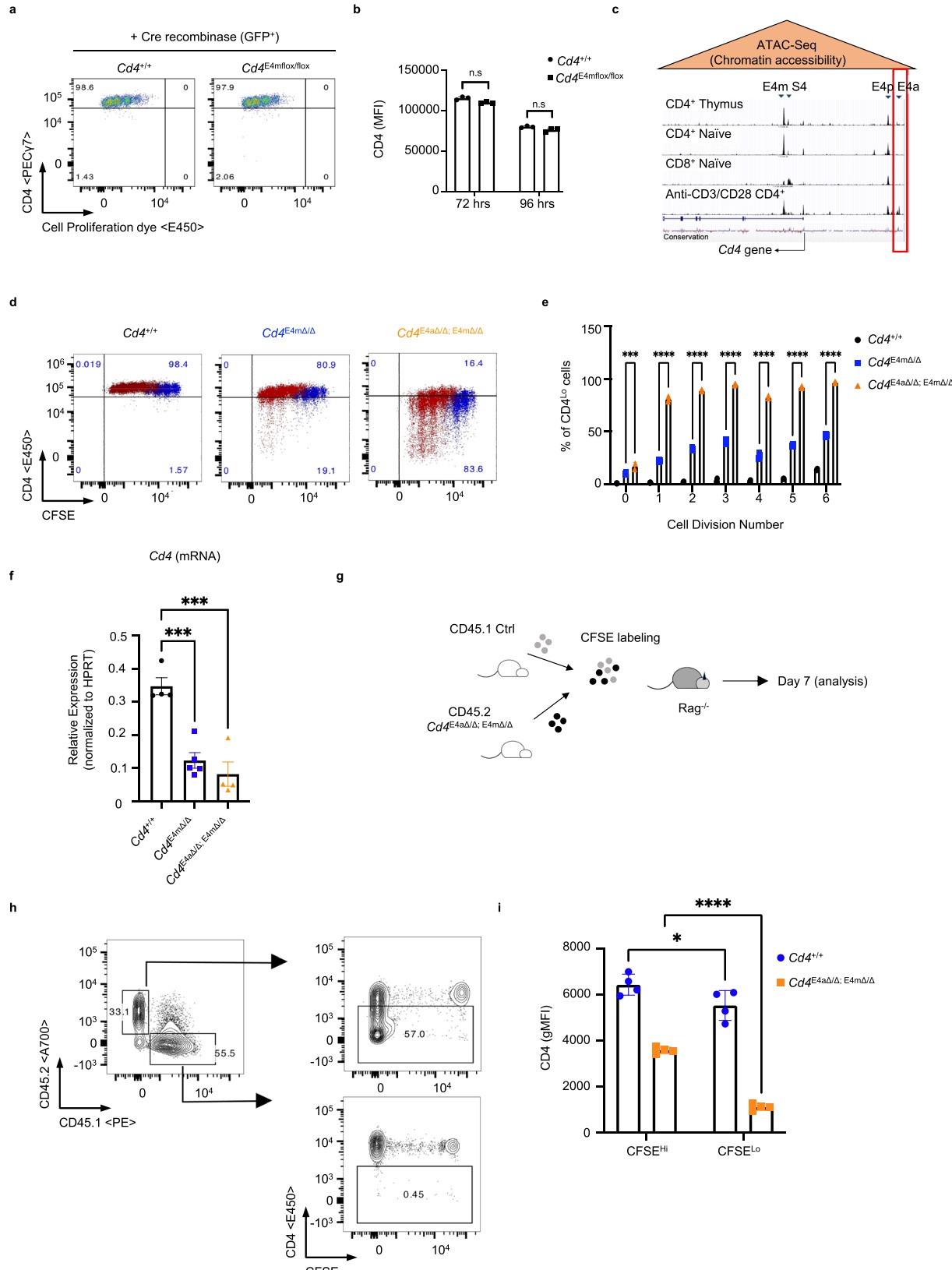

results suggest that E4a modulates CD4 expression in peripheral and proliferating CD4[+] T cells (Fig. 1d, Supplementary Fig. 1c). In support of this interpretation, ChIP-Seq analysis revealed the presence of both H3K27Ac and p300, which mark enhancer-like regions[10], within the E4a region in activated CD4[+] T cells (Fig. 2f) but not in DP[+] and CD4[+] thymocytes (Fig. 2g). In contrast,

H3K27Ac marks were highly enriched at the E4p and E4m regions in DP[+] and CD4[+] thymocytes, respectively (Fig. 2g), in agreement with their established enhancer activities during development[5,7]. Notably, there was enrichment of H3K4me1 marks, which demarcate primed and poised enhancers[10], at the E4a region in DP[+] and CD4[+] thymocytes, but not in activated CD4[+] T cells (Fig. 2f, g),

**Fig. 1 E4a modulates *Cd4* expression in effector T cells in a partially redundant manner with E4m.** **a** Flow-cytometry plot of CD4 expression in control and $Cd4^{E4mflox/flox}$ T cells, transduced with Cre recombinase and labeled with a cell-proliferation dye prior to activation. Transduced cells were gated on GFP. Data are representative to 3 independent experiments. **b** CD4 MFI of cells from (**a**) measured at two different time-points post transduction ($n = 3$ independent samples) and data shown are representative of three independent experiments. **c** Integrative Genome View (IGV) browser shot of ATAC-Seq data depicting chromatin accessibility within and upstream of the *Cd4* locus. Tracks shown are set at the same scale. Arrow indicate the annotated *Cd4* TSS and red box denotes the location of E4a. **d** FACS plot showing CD4 expression in proliferating T cells that were CFSE-labeled prior to in vitro activation with anti-CD3/CD28. Dot plots at 72hrs (blue dots) and 96hrs (red dots) post activation were overlaid. Data are representative of >3 independent experiments. **e** Percent of cells with low CD4 expression at indicated proliferation cycles assessed by CFSE labeling ($n = 3$ independent samples for $Cd4^{E4m\Delta/\Delta}$ and $Cd4^{E4m\Delta/\Delta/\ E4a\Delta/\Delta}$). Data shown are representative >3 experiments. ***$p = 0.0003$, ****$p < 0.0001$ (two-way ANOVA with Sidak Multiple Comparisons test). **f** CD4 mRNA expression (exon 3-4) in activated CD4 T cells. RNA was isolated 96hrs post activation with anti-CD3/CD28 ($n = 4$ for group 1 and 3; $n = 5$ for group 2). Data is expressed as mean ± SEM. ***$p = 0.004$, $p = 0.0002$ (one-way ANOVA and Dunnett's test). **g** Schematic illustration of experimental design. Naive WT CD45.1 T cells or CD45.2 T cells from $Cd4^{E4m\Delta/\Delta/\ E4a\Delta/\Delta}$ mice were FACS-sorted, CFSE labeled and mixed at a 1:1 ratio before i.v. transfer into $Rag^{-/-}$ recipients. Seven days later, LNs and spleen were isolated and analyzed by flow cytometry. **h** FACS plot showing CD4 expression in CD4 T cells from $Rag^{-/-}$ mice at day 7 post transfer. Cells were gated on congenic markers to distinguish WT and mutant T cells. Data are representative of two independent experiments. **i**, CD4 geometric MFI in cells gated on CSFE staining profiles ($n = 3$ independent samples for $Cd4^{+/+}$ and $n = 5$ independent samples for $Cd4^{E4m\Delta/\Delta/\ E4a\Delta/\Delta}$). Data are expressed as mean ± SEM and is representative of two independent experiments. *$p = 0.0174$, ****$p < 0.0001$ (two-way ANOVA with Bonferroni multiple comparison test).

suggesting that while E4a is not active and not required during development of CD4-lineage T cells, it is primed early during thymic differentiation.

**Lack of DNA demethylation during development affects the function of E4m/E4a in effector CD4-lineage T cells.** The role of intragenic methylation in the regulation of *Cd4* gene expression remains paradoxical. Tet1/3 cDKO naive helper T cells (*Rorc(t) Cre*Tg *Tet1/3*fl/fl) have similar expression of CD4 compared with WT naive T cells, despite significantly higher intragenic DNA methylation[5]. However, Tet1/3cDKO helper T cells have substantial loss of CD4 expression upon proliferation, similarly to $Cd4^{E4m\Delta/\Delta}$ mice[5]. We thus hypothesized that DNA methylation compromises the function of E4m/E4a in effector CD4-lineage T cells, potentially explaining the need for demethylation during development. Notably, there are no CpG motifs in the E4a CRE. However, DNA methylation marks, determined by CATCH-Seq as previously described[4,5,11], were present in regions flanking the E4a site as well as the E4m region in naive Tet1/3 cDKO compared with naive WT controls, (Fig. 3a, Supplementary Fig. 3a, b). Furthermore, we found a significant reduction in H3K4me3 at the promoter region of *Cd4* (Fig. 3b) and a significant gain of H3K9me3 at the E4a region in Tet1/3cDKO naive helper T cells compared with controls (Fig. 3c), suggestive of a less-permissive chromatin environment. Activated Tet1/3cDKO helper T cells also displayed reduced H3K4me3 at the *Cd4* promoter and elevated H3K9me3 at the E4a region, concordant with reduced function of E4a during T cell proliferation (Supplementary Fig. 3c, d). Since the defect in CD4 expression was less profound in Tet1/3cDKO compared with $Cd4^{E4m\Delta/\Delta/\ E4a\Delta/\Delta}$ effector T cells (Supplementary Fig. 3e), we reasoned that all enhancer activity was lost in $Cd4^{E4m\Delta/\Delta/\ E4a\Delta/\Delta}$ effector T cells, whereas enhancer function was only partially impaired in Tet1/3cDKO T cells. To test this, T cells were treated with an inhibitor to the histone acetyltransferases p300/cAMP response-element binding protein (CBP) which are key transcriptional activators recruited to enhancers and important for their function[12–14]. In agreement with our hypothesis, inhibition of p300/CBP activity reduced CD4 expression in a dose-dependent manner in control and Tet1/3cDKO T cells, but had no impact on $Cd4^{E4m\Delta/\Delta/\ E4a\Delta/\Delta}$ T cells (Supplementary Fig. 3e). Next, we examined DNA methylation at the *Cd4* locus in $Cd4^{E4a\Delta/\Delta}$ and $Cd4^{E4m\Delta/\Delta/\ E4a\Delta/\Delta}$ activated effector T cells. E4a deletion resulted in modestly increased DNA methylation downstream of the *Cd4* TSS and in the upstream intergenic region (Fig. 3d, e). In contrast, activated $Cd4^{E4m\Delta/\Delta/\ E4a\Delta/\Delta}$ CD4 T cells displayed hyper DNA methylation both in the *Cd4*

promoter and downstream of the TSS (Fig. 3e), which correlated with the dramatic loss of CD4 in these cells. Therefore, we surmise that DNA demethylation during thymic development plays an important role in shaping the epigenetic landscape for optimal regulation of *Cd4* expression in effector CD4-lineage T cells.

**Reduced enhancer activity as a result of DNA methylation leads to *Cd4* promoter silencing during replication of effector CD4+ T cells.** We next postulated that hypermethylation of the *Cd4* promoter in activated $Cd4^{E4m\Delta/\Delta/\ E4a\Delta/\Delta}$ T cells is the cause of loss of *Cd4* expression and is a consequence of the reduction in transcriptional activity directed by E4m and E4a. We therefore tested whether boosting enhancer activity prevents DNA methylation at the promoter and restores *Cd4* expression. To do so, we took advantage of the transcription factor TCF1, which we previously demonstrated binds to the CRE E4p[5]. We had shown that overexpression of β-catenin, a co-activator of TCF-1, in activated $Cd4^{E4m\Delta/\Delta}$ T cells, fully restored CD4 expression in the absence of E4m by maintaining E4p-mediated transcription[5] (Fig. 4a). However, overexpression of β-catenin in $Cd4^{E4m\Delta/\Delta/\ E4a\Delta/\Delta}$ T cells led to only a partial rescue of CD4 expression (Fig. 4b, c), while CD4 expression was restored to control levels in $Cd4^{E4m\Delta/\Delta}$ and Tet1/3cDKO T cells (Fig. 4c, d), suggesting that E4p activity is not sufficient to rescue CD4 expression in effector cells lacking both E4m and E4a. To test whether rescue of CD4 expression in $Cd4^{E4m\Delta/\Delta}$ T cells by TCF1/β-catenin was a consequence of changes in DNA methylation, we examined $Cd4^{E4m\Delta/\Delta}$ T cells that lost CD4 expression during replication (CD4Lo) and β-catenin-transduced $Cd4^{E4m\Delta/\Delta}$ CD4 T, whereby CD4 expression was fully restored (CD4Hi). CD4Hi $Cd4^{E4m\Delta/\Delta}$ T cells displayed a remarkable absence of DNA methylation proximal to the *Cd4* TSS, while intragenic methylation was unchanged (Fig. 4e), indicating that β-catenin likely prevents the establishment of de novo methylation at the promoter, which causes a loss of CD4 expression. These results also suggest that methylation in the intragenic region of *Cd4* does not directly impede gene expression as it was unchanged when CD4 expression was restored.

We next assessed whether gain of methylation at the *Cd4* promoter during replication was due to activity of the maintenance methyltransferase DNMT1 and/or the de novo methyltransferase DNMT3a[15]. To exclude a confounding role on gene regulation by the silencing element S4, which mediates suppression of CD4 in cytotoxic CD8 T cells[16–18], *Rorc(t)Cre*Tg $Cd4^{S4\ flox/flox}$ mice (which lack both E4m and S4)[8,18] were bred to *Dnmt3a*flox/flox mice. In the absence of DNMT3a, we observed a

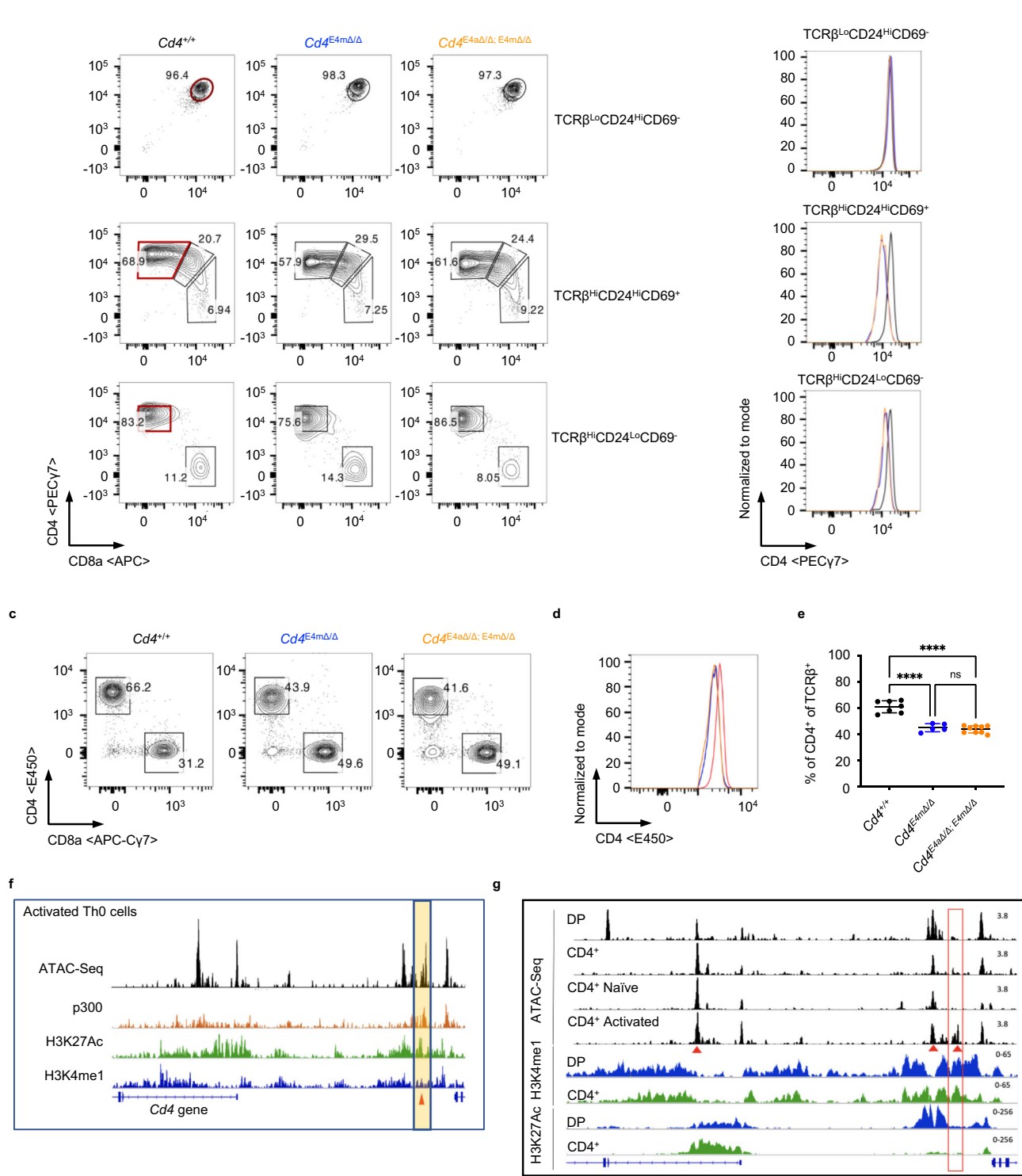

significantly lower proportion of T cells with reduced CD4 expression during replication, as well as an increase in expression of CD4 compared with controls (Supplementary Fig. 4a–c). In addition, shRNA knockdown of *Dnmt1* in *Rorc(t)Cre*[Tg] *Cd4*[S4 flox/flox]; *Dnmt3a*[flox/flox] T cells led to further rescue in CD4 expression (Supplementary Fig. 4d), suggesting that both DNMT1 and DNMT3a have roles in *Cd4* silencing during replication. Knockdown of *Dnmt1* in *Cd4*[E4mΔ/Δ/ E4aΔ/Δ] T cells led to a modest rescue of CD4 expression (Fig. 4f), supporting a partial contribution of DNA methylation to the silencing of

*Cd4* in the absence of E4m and E4a, although it is noteworthy that only a very modest rescue was achievable in activated *Cd4*[E4mΔ/Δ/ E4aΔ/Δ] T cells due to the timing of shRNA knockdown. To elucidate how β-catenin prevented de novo methylation, we next assessed H3K4me3 levels. Compared with control-vector-transduced T cells, β-catenin-transduced *Cd4*[E4mΔ/Δ] T cells had higher enrichment of H3K4me3 marks at the promoter and TSS sites (Fig. 4g), suggesting that H3K4me3 prevents the gain of new DNA methylation, in support of the antagonistic relationship between H3K4me3 and DNA

**Fig. 2 E4a is a stimulus-responsive *cis*-regulatory element licensed during development. a–c** Flow-cytometry analysis of T lymphocytes from $Cd4^{+/+}$, $Cd4^{E4m\Delta/\Delta}$, and $Cd4^{E4m\Delta/\Delta/\ E4a\Delta/\Delta}$ mice. **a** FACS contour plots showing preselected TCRβ[lo]CD69[-]CD24[hi]DP (top panel), recently selected TCRβ[hi]CD69[+]CD24[hi] (middle panel), and mature TCRβ[hi]CD69[-]CD24[lo] T-cell populations (bottom panel) in the thymus. **b** Histogram overlay of CD4 expression gated on CD4 populations at different stages in maturation denoted in red boxes in (**a**). **c** FACS contour plots showing CD4 and CD8 proportions among TCRβ + T cells from the spleen/LN. **d** Histogram overlay of CD4 expression gated on TCRβ + CD4[+] T cells from the spleen/LN of mice. Data are representative of >3 experiments. **e** Frequency of CD4[+] T cells among TCRβ + T cells in the spleen/LN of mice with indicated genotypes ($n = 7$ for $Cd4^{+/+}$, $n = 5$ for $Cd4^{E4m\Delta/\Delta}$ and $n = 9$ for $Cd4^{E4m\Delta/\Delta/\ E4a\Delta/\Delta}$). Data are expressed as mean ± SEM. ****$p < 0.0001$, ns not significant, $p = 0.8066$ (one-way ANOVA and Tukey's multiple-comparison test). **f** IGV snapshots of the *Cd4* locus in activated T (Th0) cells depicting accessibility sites by ATAC-Seq, p300 binding, and presence of H3K27Ac and H3K4me1 marks by ChIP-Seq. The highlighted yellow box and arrow indicates the location of E4a. **g** IGV snapshots of the *Cd4* locus depicting accessibility sites by ATAC-Seq in thymocytes and activated T cells, and H3K27Ac and H3K4me1 signatures by ChIP-Seq in DP and CD4 + thymocytes. Red box indicates the location of E4a and red arrow denotes the location of E4m, E4p, and E4a.

methylation[19,20]. Together, these findings suggest that a lack of thymic DNA demethylation impairs the activity of E4a and E4m in effector T cells, which is required to maintain high H3K4me3 at the *Cd4* promoter during replication, in order to repel de novo DNA methylation that leads to *Cd4* gene silencing (Supplementary Fig. 4e).

**Reduced CD4 expression in effector T cells impairs parasitic clearance during Leishmaniasis.** The need for continuous regulation of the *Cd4* gene in proliferating T cells brought into focus the importance of CD4 expression in effector T cells. Indeed, while CD4 coreceptor levels can vary in T-cell subsets in vitro[21], whether differential CD4 levels have a role in dictating cell fate of effector CD4 T cells in vivo is unclear. However, TCR signaling has been implicated in dictating the fate of differentiating effector T cells during infection[22,23]. We therefore tested the biological impact of reduced CD4 expression in a *Leishmania major* (*L. major*) infection model that induces a canonical Th1 response in WT C57BL/6 mice[24]. Footpads of $Cd4^{+/+}$ and $Cd4^{E4m\Delta/\Delta/\ E4a\Delta/\Delta}$ mice were cutaneously infected with *L. major* and swelling was assessed over time. Interestingly, while all animals developed footpad swelling, $Cd4^{E4m\Delta/\Delta/\ E4a\Delta/\Delta}$ mice displayed delayed swelling relative to $Cd4^{+/+}$ mice. However, inflamed lesions ultimately resolved at the same time in $Cd4^{+/+}$ and $Cd4^{E4m\Delta/\Delta/\ E4a\Delta/\Delta}$ mice (Fig. 5a). In contrast, no difference in footpad swelling was seen in $Cd4^{E4m\Delta/\Delta}$ or $Cd^{E4a\Delta/\Delta}$ mice relative to $Cd4^{+/+}$ mice (Fig. 5b, c). However, $Cd4^{E4m\Delta/\Delta/\ E4a\Delta/\Delta}$ mice showed significantly elevated parasitic burden in the footpad relative to controls, both at the peak of infection (d28) and at the resolution of inflammation (d54), but not early in infection (d9), suggesting that delayed inflammatory swelling in $Cd4^{E4m\Delta/\Delta/\ E4a\Delta/\Delta}$ correlates with impaired parasitic clearance (Fig. 5d–f). Activated CD44[hi] T cells from the draining lymph nodes (dLNs) of infected $Cd4^{E4m\Delta/\Delta/\ E4a\Delta/\Delta}$ mice displayed a dramatic downregulation of CD4 relative to $Cd4^{+/+}$ or $Cd4^{E4m\Delta/\Delta}$ and $Cd^{E4a\Delta/\Delta}$ mice at day 9 (Fig. 5g, h, Supplementary Fig. 5a–c). Furthermore, the percentage of cells with reduced CD4 expression in $Cd4^{E4m\Delta/\Delta/\ E4a\Delta/\Delta}$ mice significantly increased over time (Fig. 5i), recapitulating the phenotypic loss of CD4 expression upon proliferation seen in vitro and in $Rag^{-/-}$ mice (Fig. 1d, h). Together, these findings suggest that compromised TCR signaling due to unstable CD4 expression in helper T cells during *L. major* infection results in delayed inflammation and defective parasite clearance.

**Reduced CD4 expression impairs the differentiation of Th1 cells.** We next examined whether loss of CD4 impacts the antigen-specific response to *L. major* by assessing the proportion of CD11a[hi]CD49d[+] and CD11a[hi]CD44[+] T cells in the dLNs of *L. major*-infected mice. CD11a in combination with CD49d or CD44 is routinely used as surrogate marker to track antigen-experienced CD4[+] T cells in viral and parasitic infections[25–28]. At day 9 p.i., the

proportion of both CD11a[hi]CD49d[+] and CD11a[hi]CD44[+] T cells in the dLNs of *L. major*-infected mice was significantly higher than in uninfected controls (Fig. 6a, b, Supplementary Fig. 6a). No significant differences were observed in the proportion of these cells in $Cd4^{+/+}$ compared with $Cd4^{E4m\Delta/\Delta/\ E4a\Delta/\Delta}$ mice, suggesting that reduced CD4 expression did not affect the ability of T cells to engage TCR signaling. However, antigen-experienced $Cd4^{E4m\Delta/\Delta/\ E4a\Delta/\Delta}$ T cells had lower expression of CD5 and Nur77, suggestive of reduced TCR-signal strength (Fig. 6c, d)[29,30]. At 28 days post infection (d.p.i.), $Cd4^{E4m\Delta/\Delta/\ E4a\Delta/\Delta}$ mice had significantly lower frequency and number of Tbet[+]CD11a[hi]CD44[+] helper T cells in the dLNs compared with $Cd4^{+/+}$ controls (Fig. 6e–g). However, no significant difference was observed in the frequency of GATA3[+] CD11a[hi]CD44[+] T cells, which were only weakly induced independent of genotype (Supplementary Fig. 6b). Of note, no statistical difference in the absolute T cell numbers in the dLNs of d28 infected $Cd4^{+/+}$ and $Cd4^{E4m\Delta/\Delta/\ E4a\Delta/\Delta}$ mice was observed (Supplementary Fig. 6c). Since a transient IL-4-producing population has been observed early during infection with *L. major* in C57BL/6 mice[31], we examined effector T cells in the dLNs at 9 d.p.i. However, we did not observe a skewed GATA3[+] CD11a[hi]CD44[+] population in $Cd4^{E4m\Delta/\Delta/\ E4a\Delta/\Delta}$ mice nor an increase in IL-4 levels in the footpads of infected mice (Supplementary Fig. 6d, e). Similarly, in an acute LCMV infection that also drives Th1 differentiation in mice[32], we found equivalent proportions of antigen-experienced T cells in $Cd4^{+/+}$ and $Cd4^{E4m\Delta/\Delta/\ E4a\Delta/\Delta}$ mice 8 d.p.i (Supplementary Fig. 6f), but significantly lower numbers of Tbet[+] helper T cells in the spleen of $Cd4^{E4m\Delta/\Delta/\ E4a\Delta/\Delta}$ mice (Supplementary Fig. 6g). No statistical difference in GATA3[+] helper cells was detected (Supplementary Fig. 6h). Concordantly, IFN-γ expression was significantly reduced in $Cd4^{E4m\Delta/\Delta/\ E4a\Delta/\Delta}$ T cells upon restimulation with the LCMV peptide GP$_{61-80}$ (Supplementary Fig. 6i). Of note, absence of CD4 did not influence anti-CD3/CD28-mediated Th1 or Th2 in vitro differentiation (Supplementary Fig. 6j, k). To probe whether TCR–pMHC binding was compromised when CD4 expression was downregulated, we employed tetramer binding to the dominant MHC-class II-restricted glycoprotein 66–77 (GP 66–77) epitope of LCMV[33]. Remarkably, reduced CD4 expression led to no significant difference in the proportion of GP$_{66-77}$ + CD4[+] cells in $Cd4^{E4m\Delta/\Delta/\ E4a\Delta/\Delta}$ mice (Fig. 6h, i), although a complete loss of CD4 resulted in diminished proportions (Fig. 6j). These results are consistent with the notion that the CD4 coreceptor plays a marginal role in increasing the avidity of TCR–pMHC binding[34,35] and likely influences Th1 differentiation by modulating TCR signaling via LCK recruitment[36] and/or by increasing the duration of pMHC–TCR interactions[37].

**TET-mediated demethylation during thymic development is critical for optimal gene function in effector T cells.** To identify genes that share common regulatory features with *Cd4*, we next employed a genome-wide analysis using publicly available

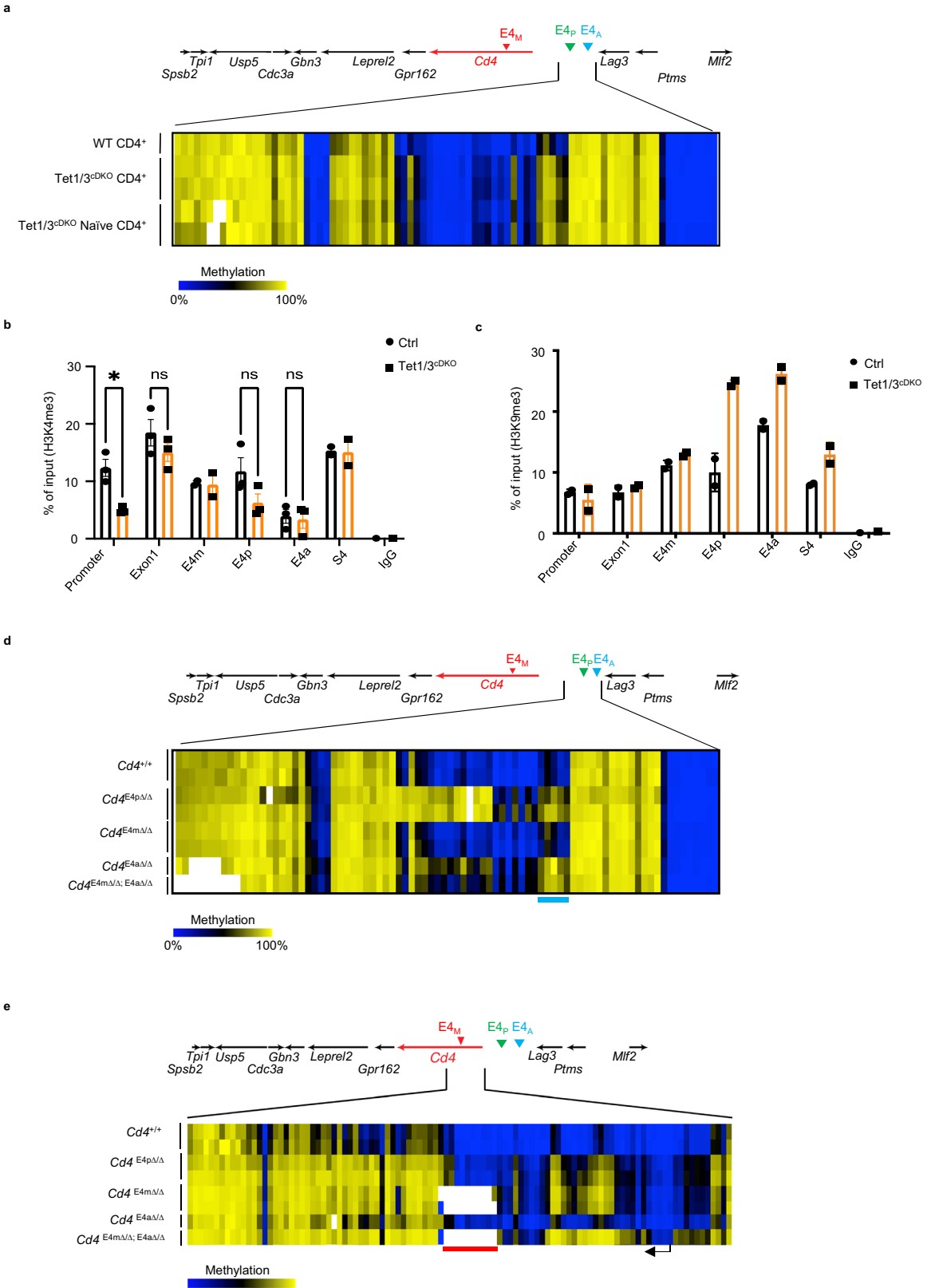

datasets for ATAC-Seq, H3K27Ac ChIP-Seq, RNA-Seq, and global 5hmC profiling by cytosine-5-methylenesulfonate immunoprecipitation (CMS-IP) to assemble a list of genes expressed in developing DP and CD4⁺ thymocytes with putative CREs that undergo DNA demethylation. We hypothesized that CREs active during thymic development would be critical in coordinating

DNA demethylation and that lack of demethylation would impair gene expression in effector T cells by impairing the activity of stimulus-responsive CREs.

Of the 6719 genes that underwent a fold change >2 in gene expression from DN3 to CD4 single-positive (SP) T-cell specification in the thymus, 409 genes (10.8%) demonstrated

**Fig. 3 Lack of DNA demethylation during development affects the function of E4m/E4a in effector CD4-lineage T cells. a** Heatmap depicting percent CpG methylation in control CD4+ (Tet1/3$^{flox/flox}$), Tet1/3$^{cDKO}$ CD4+ mature thymocytes and Tet1/3$^{cDKO}$ CD4+ naive peripheral T cells for CpGs −9270bp to −15869bp relative to the *Cd4* TSS (Chr6:124847307–124853906; mm9). The blue line underlines CpGs flanking the E4a region. CATCH-seq was performed on genomic DNA from sorted populations of TCRβ$^{hi}$CD24$^{lo}$CD69$^-$CD4$^+$CD8$^-$ thymocytes or CD4$^+$TCRβ$^+$CD62L$^{hi}$ CD44$^-$ T cells from LN/spleen. Replicates are from two independent mice. **b** H3K4me3 modifications assessed by ChIP-qPCR in sorted naive CD4 T cells. $n = 2$ or 3 mice/ genotype. Data shown are a summary of two independent experiments and expressed as mean ± SEM. *$p = 0.0257$, ns not significant, $p = 0.9797$, $p = 0.187$, and $p > 0.9999$ (two-way ANOVA and Bonferroni test). **c** H3K9me3 modifications assessed by ChIP-qPCR in sorted naive CD4 T cells. $n = 2$ mice/genotype and representative of two independent experiments. **d** Heatmap depicting percent CpG methylation in activated (WT) *Cd4*$^{+/+}$, *Cd4*$^{E4pΔ/Δ}$, *Cd4*$^{E4mΔ/Δ}$,*Cd4*$^{E4aΔ/Δ}$, and *Cd4*$^{E4mΔ/Δ/ E4aΔ/Δ}$ T cells for CpGs -9270 bp to -15869 bp relative to the *Cd4* TSS (Chr6:124847307-124853906; mm9). The blue line underlines CpGs flanking the E4a region. CATCH-seq was performed on genomic DNA from naive T cells activated in vitro with anti-CD3/CD28 for 120 hrs. Note that data for *Cd4*$^{E4pΔ/Δ}$ and *Cd4*$^{E4mΔ/Δ}$ conditions were from previously published experiments with similar experimental conditions[4,5]. **e** Heatmap depicting percent CpG methylation in WT *Cd4*$^{+/+}$, *Cd4*$^{E4pΔ/Δ}$, *Cd4*$^{E4mΔ/Δ}$,*Cd4*$^{E4aΔ/Δ}$, and *Cd4*$^{E4mΔ/Δ/ E4aΔ/Δ}$ T cells for CpGs from +6200 to −669 relative to the *Cd4* TSS (Chr6:124832027–124838896; mm9). A red line underlines CpGs in E4m (indicated by the gap in the mutant mice) and a black arrow indicates the *Cd4* TSS. Red box indicates the *Cd4* promoter. Note that data for *Cd4*$^{E4pΔ/Δ}$ and *Cd4*$^{E4mΔ/Δ}$ conditions were from previously published experiments with similar experimental conditions[4,5].

active demethylation in CD4$^+$ T cells, as assessed by the presence of 5hmC detected via CMS-IP by Tsagaratou et al.[2] (Supplementary Fig. 7a). About 81.4% of those genes (333 genes) showed a positive correlation with increased gene expression in CD4 SP cells and presence of 5hmC (Supplementary Fig. 7a, Supplementary Data 1). Interestingly, a fraction of these genes (24%–99 genes) displayed higher levels of 5hmC marks in DP precursors, while most genes showed a further gain in 5hmC in CD4 SP T cells (Supplementary Fig. 7b, Supplementary Data 2), suggesting that the demethylation process is initiated early during commitment to the CD4 lineage, congruent with our previous results[4]. Of the 350 genes that upregulated gene expression from DN3 to CD4 SP T cells (fold change >2) and exhibited 5hmC in DP or CD4 SP thymic T cells, a striking 94% (330 genes) had open-chromatin peaks and concomitant H3K27Ac marks present in their gene bodies or upstream of their annotated TSS, highlighting an intimate link between CRE activity and presence of 5hmC (Fig. 7a, Supplementary Data 3).

We next manually curated the list of 350 genes to identify genes that contain new accessible peaks when activated in vitro (Th0 cells), similarly to *Cd4*. Strikingly, 44% among those genes had novel accessible peaks downstream or upstream of their annotated TSS in activated CD4$^+$ T cells (Supplementary Fig. 7c, Supplementary Data 4). Gene ontology and enrichment analysis revealed that a significant proportion of these genes had fundamental roles in regulating TCR activation, signaling and differentiation pathways (Fig. 7b). Indeed, many of these genes such as *Cd5*, *Cd6*, and *Zbtb7b* have been shown to modulate critical post-thymic functions in helper T cells[38–42], suggesting that their epigenetic programming during development may be critical for their optimal expression in effector T cells. To assess the consequence of impaired thymic DNA demethylation, we generated *Rorc(t)*$^{CreTg}$ *Tet1*$^{fl/fl}$ *Tet2*$^{fl/+}$ *Tet3*$^{fl/fl}$ mice in which excision of TET alleles occurs during the transition of immature single-positive (ISP) CD8$^+$ to DP thymocytes[43]. We reasoned that deletion of one allele of TET2 may allow for recovery of genes that undergo demethylation in a TET2-dependent manner, but without impairing homeostasis, as akin to *Cd4*$^{CreTg}$ *Tet 2/3* floxed mice, *Rorc(t)*$^{CreTg}$ *Tet1/2/3* compound floxed mice have profound developmental defects, display severe autoimmune phenotypes with an onset of 4–5 weeks, and enlarged spleens and lymph nodes[44] (Supplementary Fig. 7d), making the assessment of post-thymic effector T cells difficult. In contrast, *Rorc(t)*$^{CreTg}$ *Tet1*$^{fl/fl}$ *Tet2*$^{fl/+}$ *Tet3*$^{fl/fl}$ mice were healthy, displayed no splenomegaly or lympho-adenopathies and had equivalent proportions of naive CD44$^-$CD62L$^+$ T lymphocyte compartments as control mice (Supplementary Fig. 7e). Furthermore, naive CD4 T cells from *Rorc(t)*$^{CreTg}$ *Tet1*$^{fl/fl}$ *Tet2*$^{fl/+}$ *Tet3*$^{fl/fl}$ mice

had comparable CD4 expression to controls (Supplementary Fig. 7f). However, upon T cell proliferation, CD4 expression was dramatically reduced (Fig. 7c, Supplementary Fig. 7g). In a similar pattern, ThPOK (*Zbtb7b* gene), CD5 and CD6 protein and mRNA expression were significantly reduced following proliferation (Fig. 7c, d, Supplementary Fig. 7g). Upon cotransfer of naive *Rorc(t)*$^{CreTg}$ *Tet1*$^{fl/fl}$ *Tet2*$^{fl/+}$ *Tet3*$^{fl/fl}$ and CD45.1 CD4 T cells into *Rag*$^{-/-}$ mice, there was significant downregulation of CD5, CD6, and ThPOK expression at 7 days post transfer (Supplementary Fig. 7h). Of note, a reduction in CD5 and CD6 expression on naive T cells was also seen prior to activation (Supplementary Fig. 7f), suggesting a role for DNA demethylation in the regulation of these genes in the periphery. Nonetheless, these were significantly further downregulated following replication (Fig. 7c, d, Supplementary Fig. 7g). *Cd5*, *Cd6*, and *Zbtb7b* loci displayed novel chromatin accessible sites in activated CD4$^+$ T cells (red arrows), which coincided with the presence of H3K27Ac and/or presence of Pol-II marks, suggestive of putative CREs (Fig. 7e–g). Together, our findings highlight the conservation of similar patterns of epigenetic programming as *Cd4* in a cohort of genes and demonstrate the importance of DNA demethylation during development in ensuring proper gene function in effector helper T cells.

## Discussion

In this study, using the *Cd4* gene as a model locus, we have demonstrated mechanistically how DNA methylation in non-promoter regions affects gene expression and illustrated the biological importance of DNA demethylation early during thymic development.

DNA demethylation during T-cell development sculpted the chromatin landscape for the subsequent function of stimulus-responsive regulatory elements in peripheral effector T cells. We showed that a novel stimulus-responsive CRE, E4a, acted in a partially redundant manner with a developmental CRE, E4m, to maintain CD4 expression in effector T cells. In the absence of TET1/3-mediated DNA demethylation in the thymus, the functions of E4a/E4m were compromised in effector cells and resulted in a loss of CD4 expression during cell proliferation. This loss strongly correlated with de novo DNA methylation at the *Cd4* promoter and gene expression could be partially rescued by knocking down *Dnmt1* and *Dnmt3a*. De novo methylation was a result of reduced transcriptional activity and reduced promoter H3K4me3 levels, as boosting enhancer function restored CD4 expression during replication and prevented gain of new methylation at the *Cd4* promoter by maintaining H3K4me3. Together, these findings highlight a novel role for DNA demethylation in the licensing of stimulus-responsive elements early

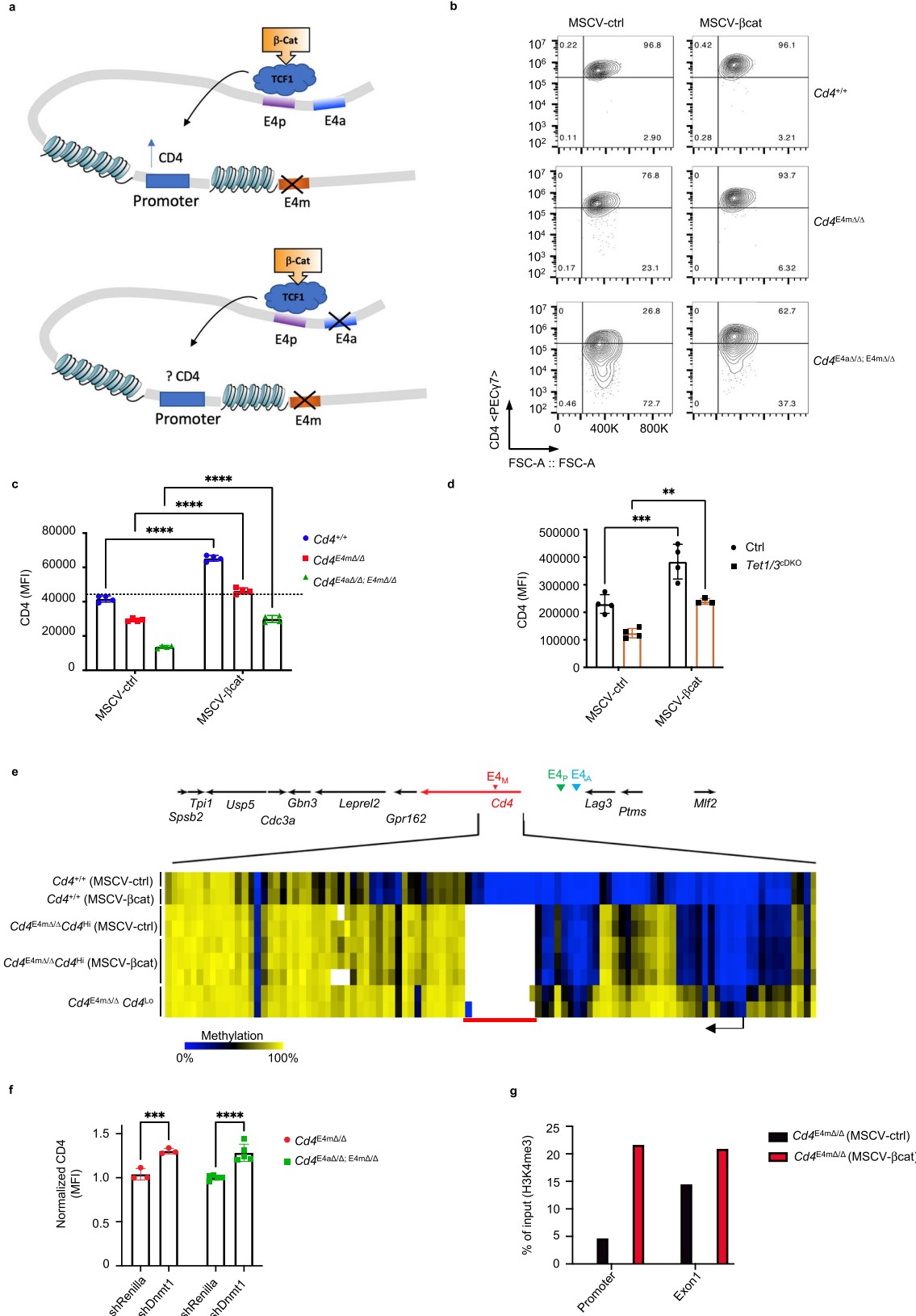

during development to impart a heritable gene-expression program that is critical in preventing spurious de novo DNA methylation in replicating effector T cells.

Our findings are in agreement with the established antagonistic relationship between H3K4me3 and de novo DNA methylation[20,45,46]. In a previous study using a GAL-4/ TATA-binding protein-tethering system, it was elegantly shown that the positioning of the transcription machinery before implantation determines what regions will be protected from subsequent de novo methylation in blastocysts[47]. Transcription factors and the RNA-polymerase complex were shown to play a major role in protecting recognized regions from de novo

**Fig. 4 Reduced enhancer activity as a result of DNA methylation leads to *Cd4* promoter silencing during replication of effector CD4[+] T cells.**
**a** Schematic illustration depicting the location and orientation of the *Cd4* enhancers, and TCF1 binding site previously validated[5]. **b** FACS contour plots showing expression of CD4 in activated T cells with indicated genotypes upon rescue with a control vector or a vector overexpressing β-catenin. Data shown are 96hrs post retroviral transduction and representative of >3 independent experiments. **c** CD4 gMFI on T cells analyzed 96hrs post retroviral transduction with a control vector or β-catenin vector (*n* = 4 independent samples). Data shown is representative of three experiments and expressed as mean ± SEM. ****$p < 0.0001$ (2-way ANOVA and Sidak's multiple comparison test). **d** CD4 gMFI on T cells analyzed 96hrs post retroviral transduction with a control vector or β-catenin vector (*n* = 4 independent samples). Data shown are representative of three experiments and expressed as mean ± SEM. **$p = 0.0049$, ***$p = 0.0003$ (two-way ANOVA and Sidak's multiple comparison test). **e** Heatmap depicting percent CpG methylation CpGs from +6200 to −669 relative to the *Cd4* TSS (Chr6:124832027–124838896; mm9) in activated WT CD4[+] and *Cd4[4mΔ/Δ]* CD4 + T cells transduced with a control vector or β-catenin. A red line underlines CpGs in E4m (indicated by the gap in the mutant mice) and a black arrow indicates the *Cd4* TSS. CATCH-seq was performed on genomic DNA from *Cd4[E4mΔ/Δ]* GFP[+] sorted populations (control vector *n* = 2 or β-Catenin transduced *n* = 3). Note that data for CD4[Lo]*Cd4[E4mΔ/Δ]* and *Cd4[+/+]* conditions were from previously published experiments[5]. **f** Normalized CD4 gMFI of *Cd4[E4mΔ/Δ]* and *Cd4[E4mΔ/Δ/ E4aΔ/Δ]* CD4[+] T cells transduced with a shRNA against Renilla or Dnmt1. Cells were analyzed 96hrs post transduction and gated on GFP and data were normalized to shRenilla within each group. (*n* = 3 for *Cd4[E4mΔ/Δ]*; *n* = 5 for *Cd4[E4mΔ/Δ/ E4aΔ/Δ]*) ***$p = 0.008$, ****$p < 0.0001$ (two-way ANOVA with Sidak multiple comparison test). **g** H3K4me3 modifications assessed by ChIP-qPCR in control or β-catenin transduced (GFP+) CD4[+] T cells from *Cd4[E4mΔ/Δ]* mice. Cells were FACS-sorted 96hrs post transduction. (*n* = 2 independent samples). Data are representative of three independent experiments.

methylation by recruiting the H3K4-methylation machinery[47]. Similarly, H3K4me3 was recently shown to have an instructional role in preventing repression of developmental genes via DNA methylation[48]. Our findings add to these studies, suggesting that the fidelity of maintenance of existing DNA-methylation patterns in somatic cells and/or protection against rampant de novo methylation during replication is critically dependent on H3K4me3 signatures. It is enticing to speculate that such mechanisms may be involved in some of the unknown etiologies of promoter hypermethylation of a cohort of genes in certain disorders such as cancer[49,50], although future work is needed.

While we did not find a role for E4a activity during thymic development, the intergenic space upstream of the *Cd4* locus in *Cd4[E4aΔ/Δ]* and *Cd4[E4mΔ/Δ/ E4aΔ/Δ]* T cells contained significant DNA methylation. We speculate that E4a may have a function in coordinating TET1/3-mediated demethylation together with E4m and E4p[5], or that it has a regulatory role in preventing spurious DNMT3a/3b activity in this region[51–55]. In addition, we have documented a contribution for the CD4 coreceptor levels in modulating effector T-cell differentiation. Beyond its critical role in regulating TCR signaling during thymic T-cell commitment[3,5], the importance of maintaining high CD4 expression during the differentiation of effector T cells remained unclear. We showed here that elevated CD4 expression was critical for optimal TCR signaling and promoted differentiation into Tbet-expressing Th1-lineage T cells. Our data support the idea that strength or duration of TCR signaling impacts effector T-cell differentiation, but does not skew its polarization into alternate subsets, a phenomenon likely driven by other factors[37,56]. It is noteworthy that due to the dramatic loss of CD4 during replication in *Cd4[E4mΔ/Δ/ E4aΔ/Δ]* mice, helper T cells had to be identified based on a TCRβ[+] CD8a[−] gating strategy. However, it has previously been reported that a population of MHC-class II-restricted DN T cells (TCRβ[+]CD4[−]CD8[−]) is induced during *L. major* infection[57,58]. Thus, we cannot exclude a role for CD4 in the skewing of the DN population in our model. In addition, previous studies have shown that other CD4[+] T-cell subsets, including regulatory T cells, play a role in *L. major* parasite persistence and clearance[59,60], and therefore the role of CD4 in the differentiation of these subsets in this model can also not be excluded.

Last, building on principles from genomic studies of the *Cd4* locus, we have identified a cohort of genes whose expression in effector T cells was dependent on developmental programming. Notably, the *Zbtb7b* gene shares similar characteristics with *Cd4*. *Zbtb7b* expression is modulated during positive selection and differentiation of MHC-class II-selected T cells in the thymus via

two CREs, TE and PE[61–63]. Ablation of PE and TE abolishes *Zbtb7b* expression during thymic differentiation[64]. It is not known if PE and TE coordinate the demethylation process akin to the *Cd4* CREs. Another CRE, general T cell element (GTE) situated between PE and TE, has been reported to have enhancer properties in CD4[+] T cells, although its endogenous role has not been tested[62]. Here, we have found enhanced chromatin accessibility in a putative CRE proximal to the GTE region in effector CD4[+] T cells. Given the proliferation-associated loss of expression of *Zbtb7b* in T cells deprived of Tet activity during development, we speculate that GTE may be a stimulus-responsive element sensitive to intragenic or intergenic DNA methylation. Notably, in this study, our analysis is confined to integration of genes with accessibility peaks early during early T-cell activation (Th0 conditions), whereas many genes may require additional differentiation signals for optimal expression. Future experiments are warranted to delineate CREs that modulate expression in differentiated effector subsets.

These studies will be instrumental as roughly 90% of autoimmunity-associated genetic variants identified via genome-wide association studies lie in noncoding regions of the genome. It was recently reported that stimulation-responsive chromatin regions correlate with significant trait heritability in multiple immune-cell types, pointing to potential causality of stimulus-responsive CREs in autoimmune diseases[65,66]. Thus, viewed in this perspective, our study invokes a compelling link between early developmental epigenetic processes in shaping the activity of these stimulus-responsive elements in effector T cells.

## Methods

**Mice.** F0 mice were genotyped by PCR amplification of the targeted region followed by TA cloning, and Sanger sequencing of individual colonies. Founders bearing E4a deletions in chromosome 6 with MM10 coordinates 124901815–124902199 were then backcrossed at least two times to wild-type C57BL/6 J mice. For the generation of *Cd4[E4mΔ/Δ E4aΔ/Δ]* mutant mice, *Cd4[E4aΔ/Δ]* embryos were used for CRISPR injections using the same guide RNAs used for the generation of *Cd4[E4aΔ/Δ]* mutants. The generation of *Cd4[E4mΔ/Δ]* mice has been described previously[5]. *Cd4[E4mflox/flox]* mice were also generated via CRISPR–Cas9. Briefly, two sgRNAs targeting the E4m region and donor single-stranded DNA oligos containing loxP sites with 60 bp homology to sequences on each side of the sgRNA target sites were injected into C57BL/6 J zygotes. To facilitate detection of correct insertions, the donor-template oligos were engineered to contain an Nhe1 restriction site on the 5′ arm and an EcoR1 site on the 3′ arm. Integration of loxP sites was first assessed by PCR amplification of the expected modified region followed by enzymatic digestion of PCR products by EcoR1 or NHE1. After determination of loxP sites on the same allele of E4m, the respective homologous regions bearing the loxP sites were PCR-amplified, TA-cloned, and Sanger-sequenced to ensure no mutations around integrated loxP sites.

*Tet1* and *Tet2* floxed (C57BL/6 J) and *Tet3* floxed mice (129 backcrossed to C57BL/6 J) were kindly provided by Dr. Iannis Aifantis[67] and Dr. Yi Zhang[68], respectively. These mice were then backcrossed onto RorcCre[Tg] (B6.FVB-Tg(Rorc-cre)

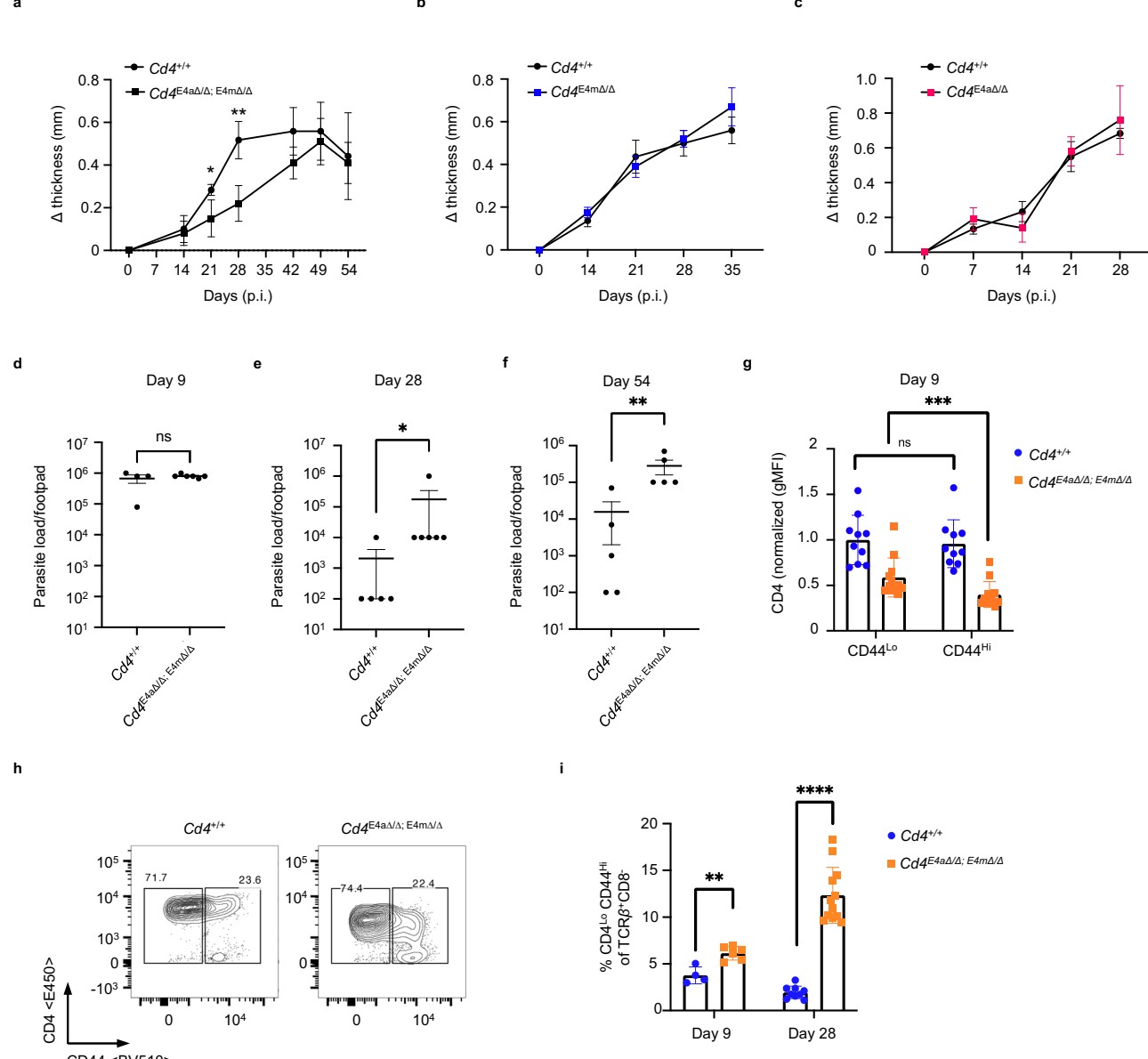

**Fig. 5 Reduced CD4 expression in effector CD4 T cells impairs parasitic clearance during Leishmaniasis. a–c** Footpads of indicated genotypes were infected with $2\times10^6$ *L. major* promastigotes. Footpad swelling was measured over days post infection (p.i.). **a** ($n = 12$ for $Cd4^{+/+}$; $n = 10$ for $Cd4^{E4m\Delta/\Delta/E4a\Delta/\Delta}$) and is a summary of three independent experiments. Data shown is mean ± SEM. *$p = 0.047619$, **$p = 0.006494$ (multiple Mann–Whitney tests, FDR = 1%) (**b**) ($n = 11$ for $Cd4^{+/+}$; $n = 10$ for $Cd4^{E4m\Delta/\Delta}$) and is a summary of two independent experiments for time points d0–d28. **c** ($n = 5$ for $Cd4^{+/+}$; $n = 9$ $Cd4^{E4a\Delta/\Delta}$) and is a summary of two independent experiments. **d–f** Parasitic burdens in the footpads of infected mice. On the indicated days p.i., mice were euthanized, and *L. major* titers in the footpads were determined using a limiting-dilution assay as described in "Methods". **d** ($n = 4$ for $Cd4^{+/+}$; $n = 6$ for $Cd4^{E4m\Delta/\Delta/E4a\Delta/\Delta}$) and is representative of two experiments. **e** ($n = 5$ for $Cd4^{+/+}$; $n = 6$ for $Cd4^{E4m\Delta/\Delta/E4a\Delta/\Delta}$) and is representative of four experiments. **f** ($n = 5$ for $Cd4^{+/+}$; $n = 5$ for $Cd4^{E4m\Delta/\Delta/E4a\Delta/\Delta}$) and is representative of 2 experiments. Data shown are mean ± SEM. ns not significant, $p = 0.9238$; *$p = 0.0130$, **$p = 0.0079$ (two-tailed Mann–Whitney U test). **g** Normalized CD4 gMFI expression on CD8⁻ TCRβ⁺ CD44Lo or CD8⁻ TCRβ⁺ CD44Hi T cells from the inguinal dLNs of infected mice 9 days p.i. ($n = 10$ for $Cd4^{+/+}$; $n = 12$ for $Cd4^{E4m\Delta/\Delta/E4a\Delta/\Delta}$) and is a summary of two independent experiments. Data shown are a mean ± SEM, ns not significant, $p = 0.7959$; ***$p = 0.000488$ (Wilcoxon matched-pairs signed rank tests). **h** FACS contour plot showing CD4 versus CD44 expression among CD8⁻ TCRβ⁺ T cells from the inguinal dLNs of infected mice 9 days p.i. Data shown are representative of two independent experiments. **i** Proportions of helper T cells with low CD4 expression among CD8⁻ TCRβ⁺ CD44Hi T cells from the inguinal dLNs of infected mice, 9 and 28 days p.i. ($n = 4$ and 9 respectively for $Cd4^{+/+}$; $n = 6$ and 12, respectively for $Cd4^{E4m\Delta/\Delta/E4a\Delta/\Delta}$) and is a summary of two independent experiments. Data shown are mean ± SEM. **$p = 0.009524$; ****$p < 0.0001$ (multiple Mann–Whitney tests, FDR = 1%).

1Litt/J #022791) mice. B6.SJL-CD45.1 mice were purchased from the Jackson Laboratory (#002014). B6.129S7-Rag1tm1Mom/J were purchased from the Jackson Laboratory (#002216). All mice were maintained under specific pathogen-free (SPF) conditions at the barrier animal facility at University of Iowa Carver College of Medicine on a 12 hour light cycle at 30–70% humidity and temperature of 20–26 degrees celsius, with access to standard chow and water. Littermate controls and sex-matched 6–8 week-old mice were used for all experiments, unless specified otherwise.

$CO_2$ exposure was used as the method for euthanasia for all experiments. Permission for all animal experiments performed was granted by the IACUC at the University of Iowa Carver College of Medicine.

**FACS sorting**. T cells were enriched from the lymph nodes and spleen of mice using the Dynabeads untouched mouse T-cell kit (Thermo Fisher Scientific) or the

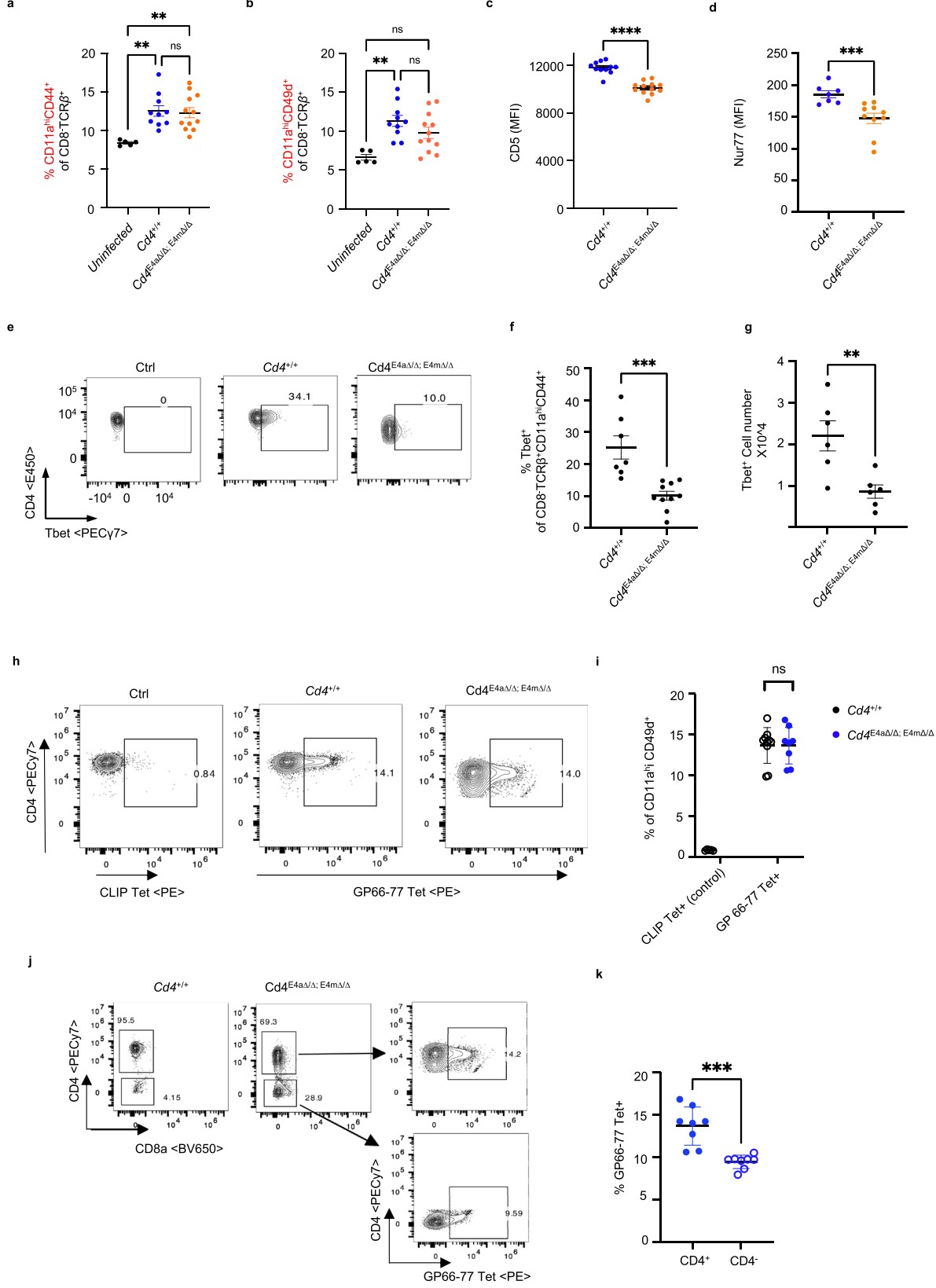

CD4+ T-cell selection kit (Miltenyi). Naive CD4+ T cells were then isolated by flow cytometry based on the markers CD4 + CD62L + CD44 − CD25 − on a FACS Aria II or FACS Fusion Sorter.

**Antibodies and flow cytometry**. The following antibodies from Tonbo, Biolegend, or ThermoFisher Scientific were used: anti-CD62L (MEL-14, 1:300), anti-CD25 (PC61.5, 1:300), anti-CD44 (IM7, 1:300), anti-CD19 (ID3, 1:300), anti-CD5 (53–7.3, 1:300), anti-CD6 (OX-129, 1:200), anti-Zbtb7b (T43–94, 1:100), anti-CD4 (RM4-5, 1:300), anti-CD8a (53–6.7, 1:300), anti-Nur77 (12.14, 1:100), anti-CD11a (M17/4, 1:300), anti-CD49d (R1–2, 1:300), anti-TNF (MP6–XT22, 1:300), anti-IFN-γ (XMG1.2, 1:300), anti-CD45.2 (104, 1:300), anti-CD45.1 (A20, 1:300), anti-GATA3 (TWAJ, 1:100), and anti-Tbet (4B10, 1:100). Ghost Dye from Tonbo (5ul/sample) was used to gate out dead cells from all analysis. LN and spleen were

**Fig. 6 Reduced CD4 expression impairs the differentiation of Th1 cells. a** Proportions of CD11a$^{hi}$CD44$^+$ T cells among CD8$^-$ TCRβ$^+$ T cells in the inguinal dLNs of mice, 9 days p.i. ($n = 5$ for uninfected; $n = 10$ for $Cd4^{+/+}$; $n = 12$ for $Cd4^{E4mΔ/Δ/E4aΔ/Δ}$). Data shown are mean ± SEM and are a summary of 2 independent experiments. ns not significant, $p > 0.9999$, **$p = 0.0047$, 0.0049 (Kruskal–Wallis test and Dunn's multiple comparison test). **b** Proportions of CD11a$^{hi}$CD49d$^+$ T cells among CD8$^-$ TCRβ$^+$ T cells in the inguinal dLNs of mice, 9 days p.i. ($n = 5$ for uninfected; $n = 10$ for $Cd4^{+/+}$; $n = 12$ for $Cd4^{E4mΔ/Δ/E4aΔ/Δ}$). Data shown are mean ± SEM and are a summary of two independent experiments. ns not significant, $p = 0.5205$ **$p = 0.0024$, *$p = 0.0376$ (Kruskal–Wallis test and Dunn's multiple comparisons test). **c** CD5 MFI on CD11a$^{hi}$CD44$^+$ T cells among CD8$^-$ TCRβ$^+$ T cells in the inguinal dLNs of mice, 9 days p.i. ($n = 11$ for $Cd4^{+/+}$; $n = 12$ for $Cd4^{E4mΔ/Δ/E4aΔ/Δ}$). Data shown are mean ± SEM. and are a summary of two independent experiments ****$p < 0.0001$ (two-tailed Mann–Whitney $U$-test). **d** Nur77 MFI on CD11a$^{hi}$CD44$^+$ T cells among CD8$^-$ TCRβ$^+$ T cells in the inguinal dLNs of mice, 9 days p.i. ($n = 7$ for $Cd4^{+/+}$; $n = 10$ for $Cd4^{E4mΔ/Δ/E4aΔ/Δ}$). Data shown are mean ± SEM and are a summary of two independent experiments, ***$p = 0.0007$ (two-tailed Mann–Whitney $U$-test). **e** Representative FACS contour plot showing Tbet expression among CD4$^+$CD8$^-$ TCRβ$^+$ T cells from the popliteal dLNs of mice, 28 days p.i. Data shown are representative of >3 independent experiments. **f** Proportion and numbers of Tbet+ cells among CD8$^-$ TCRβ$^+$ CD11a$^{hi}$CD44$^+$ T cells from the popliteal dLNs of mice, 28 days p.i. ($n = 7$ for $Cd4^{+/+}$; $n = 10$ for $Cd4^{E4mΔ/Δ/E4aΔ/Δ}$). Data shown are a summary of two independent experiments. ***$p = 0.0001$ (two-tailed Mann–Whitney U-test). **g** Numbers of Tbet+ cells among CD8$^-$ TCRβ$^+$ CD11a$^{hi}$CD44$^+$T cells from the popliteal dLNs of mice, 28 days p.i. ($n = 6$/group) **$p = 0.0087$ (two-tailed Mann–Whitney U-test). **h** Representative FACS contour plot showing GP$_{66-77}$ tetramer+ or CLIP-tetramer+ CD4 T cells (control) among splenocytes of LCMV-infected animals 8d p.i. **i** Quantification of data shown in **h** ($n = 5$ for CLIP Tet +; $n = 10$ for $Cd4^{+/+}$; $n = 8$ for $Cd4^{E4mΔ/Δ/E4aΔ/Δ}$). Data are representative of two independent experiments. ns not significant, $p = 0.8098$ (two-tailed Mann–Whitney U-test). **j** Gating strategy to detect GP$_{66-77}$ Tetramer+ cells among TCRβ + CD8- CD4 + and TCRβ + CD8- CD4- T cells. **k** Quantification of data shown in **j** ($n = 8$). Data are representative of two independent experiments. ***$p < 0.001$ (two-tailed Mann–Whitney U-test).

isolated by dissection from mice and then mashed through a 70-μm filter. Red blood cells were lysed in ACK lysis buffer (Lonza), counted, and stained in 2% IMDM (Gibco) with a Live/Dead exclusion Ghost Dye (Tonbo), followed by staining with surface markers for 30 min at 4 degrees. Details of all antibodies are provided in the supplementary method section. For transcription-factor staining, cells were fixed overnight in the Foxp3/Transcription Factor/Fixation-Concentrate kit (Tonbo) after surface staining. After fixation, cells were permeabilized and stained with the appropriate antibodies in PBS for 1 hr at room temperature. For cytokine analysis, after surface staining, cells were fixed at a final concentration of 2% PFA for 15 min at room temperature, followed by two washes with PBS and overnight storage. The next day, cells were permeabilized and stained for 1 h at room temperature for cytokines. Tetramers to MHC-class II-restricted glycoprotein 66–77 (GP$_{66-77}$) were obtained from the NIH Tetramer Core Facility and staining performed following their guidelines. Stained cells were analyzed on a Cytoflex flow analyzer or an LSR-II flow analyzer. Flow jo (v9.9.6, Tree Star) was used for all flow-cytometry analysis.

**T-cell activation and retroviral transduction.** Tissue culture plates were coated with polyclonal goat affinity-purified antibody to hamster IgG (MP Biomedical) at 37 °C for at least 2 h or overnight at 4°C and washed 2× with PBS before cell plating. FACS-sorted CD4+CD8$^-$CD25$^-$CD62L$^+$CD44$^{lo}$ naive T cells were seeded in T-cell medium [RPMI 1640 (Gibco), 10% heat-inactivated FBS (Atlanta), 2 mM L-glutamine, 50 μg/ml gentamicin, 1% Penn/Strep, and 50 uM 2-mercaptoethanol (Gibco)] on plate-bound hamster IgG along with anti-CD3 (BioXcell, clone 145-2C11, 0.25 μg/ml or Tonbo, clone 17A2, 0.25ug/mL) and anti-CD28 (BioXcell, clone 37.5.1, 1 μg/ml or Tonbo, clone 37.51, 1ug/mL) antibodies. Forty-eight hours later, cells were lifted off the plates and cultures were supplemented with 100 U/ml recombinant human IL-2 (Peprotech). CFSE labeling for certain experiments was performed prior to seeding on tissue culture plates. For Th1 and Th2 in vitro differentiation assays, FACS-sorted CD4 naive T cells were activated for 72hrs in the absence of cytokines (Th0 conditions) or Th1 conditions (50 ng/mL rIL-12 (Tonbo), 10ug/mL anti-IL-4 (Tonbo), and 50 U/mL rIL-2) or Th2 conditions (50 ng/mL IL-4, 10ug/mL anti-IFN γ (Tonbo)).

For *Dnmt1* and *Dnmt3a* knockdown, shRNA plasmids on a mIR-E backbone were provided by Johannes Zuber[69]; MSCV-beta-catenin-IRES-GFP was a gift from Tannishtha Reya (Addgene plasmid #14717). MSCV-Cre-IRES-GFP was previously described[7]. Retroviruses were packaged in PlatE cells by transient transfection using TransIT 293 (Mirus Bio) or Lipofectamine 3000 (Thermo Fisher). Cells were transduced by spin infection at 1200 × $g$ at 32 °C for 90 min in the presence of 10 μg/ml polybrene (Santa Cruz) 12–16 h post activation with anti-CD3/anti-CD28. Viral supernatants were removed the next day and replaced with fresh medium containing anti-CD3/anti-CD28. Cells were lifted off 24 h later and supplemented with 100 U/ml recombinant human IL-2.

**Methylation analysis.** Genomic DNA was isolated from FACS-sorted T-cell populations using genomic DNA isolation kits (Qiagen). Purified genomic DNA was subjected to bisulfite treatment with the Qiagen EpiTect bisulfite kit. For locus-wide bisulfite sequencing, CATCH-seq was performed as previously described[4,11] using BAC clone RP24-330J12 (BACPAC Resource Center, CHORI).

**Low-input histone ChIP-qPCR.** In total, 100,000–150,000 T cells were isolated by cell sorting and processed for ultra-low-input micrococcal nuclease-based native ChIP (ULI-NChIP)[70]. For immunoprecipitation, 2 μg of antibody was used per reaction. H3K9me3 (Lot. A2217P) and H3K4me3 (Lot No. A1052D) antibodies were from Diagenode. Immunoprecipitated and input DNA were extracted by phenol–chloroform extraction and used for qPCR with SsoAdvanced Universal SYBR Green supermix (Biorad). All primer sequences used are listed in Supplementary Table 1.

**Real-time quantitative PCR.** DNase I-treated total RNA was prepared from sorted T cells using RNeasy RNA isolation kit (Qiagen) and cDNA was synthesized using an iScript cDNA synthesis kit (Biorad). Quantitative PCR was performed using SsoAdvanced Universal SYBR Green supermix (Biorad) and a CFX Connect Real-time PCR detection system (Biorad). A list of primers used is detailed in the supplementary table. All primer sequences used are listed in Supplementary Table 1.

**Rag knockout T-cell transfer.** Sex- and age-matched CD4 + T cells were enriched from CD45.1 and naive control or RorcCre Tet1/3 mice using the Dynabeads untouched CD4 T cell kit (ThermoFisher Scientific), ensuring >85% purity. Cells were counted and CFSE labeled and mixed at a 1:1 ratio before transfer by i.v injections into 6–8-week-old *Rag-/-* recipients. In some experiments, naive CD4 T cells were FACS-sorted after enrichment with Dynabeads.

**Leishmania infection.** The *L. major* strain WHOM/IR/-173 was grown in vitro in complete M199 media supplemented with 5% HEPES, 10% FBS, and 1% penicillin–streptomycin at 26°C. Metacyclic promastigotes were isolated from stationary-phase cultures and enriched using a Ficoll gradient as described previously[71]. In-house bred and cohoused C57BL/6 J mice were used as controls. About 6–8-week-old male mice were infected with $2 × 10^6$ *L. major* promastigotes per footpad in a volume of 50 μl of PBS. Footpad measurements were taken weekly and prior to infection by caliper measurement in a single-blinded manner. For quantification of *L. major* promastigotes, footpads were surgically dissected and homogenized by douncing as previously described[71], and 10-fold serial limiting dilutions to extinction of the homogenates were plated in triplicates in 96-well flat-bottom plates in complete M199 media. Four to 6 days after culture at 26°C, each well was analyzed under a microscope for the presence or absence of *L. major* to determine the titers. Media-only controls were included to ensure no cross contamination. Promastigotes were counted in a blinded manner at the lowest dilution factor and triplicates were averaged.

**LCMV Armstrong infection.** Mice were infected with $2 × 10^5$ PFU of the Armstrong strain of LCMV via i.p injection and spleens were harvested 8 days post infection. Splenocytes from LCMV-infected mice were isolated using Collagenase D digestion and were incubated with indicated doses of the LCMV-derived peptide GP$_{66-80}$ (Anaspec) in the presence of 3 μg/ml brefeldin A (Tonbo) for 5 h at 37°C, followed by staining for IFN-γ.

**Multiplex cytokine analysis.** Homogenates collected after douncing of tissue from the footpads of infected animals were spun down to remove cellular components, and supernatants were collected and stored at -80°C for cytokine measurements using a Th1–Th2 mouse ProcartaPlex Panel (Thermo Fisher Scientific). About 50 ul of supernatants with appropriate dilutions were used for cytokine estimations and fluorescent readings were acquired on a BioRad Bio-Plex (Luminex 200) instrument.

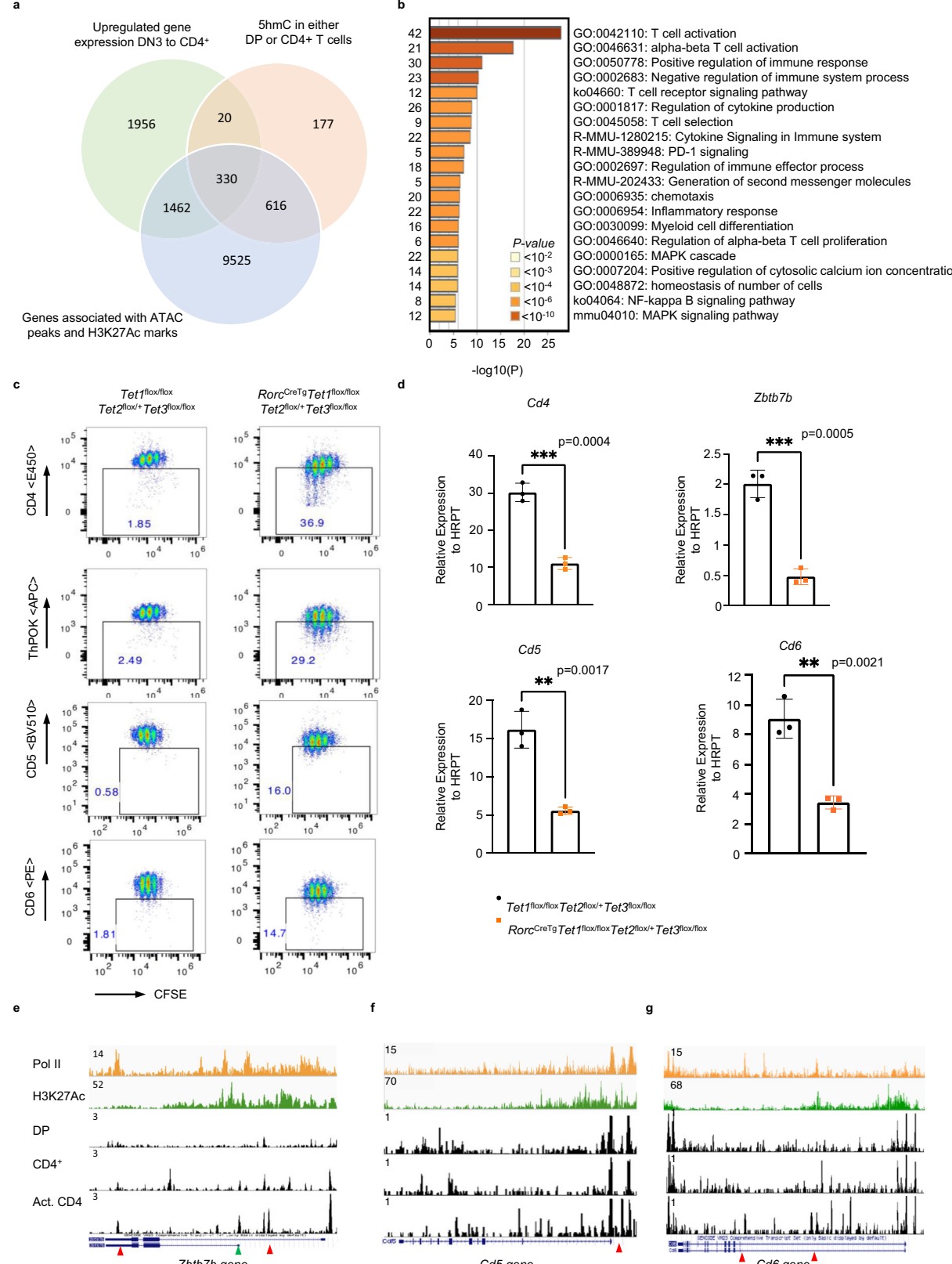

**Bioinformatics processing and analysis**. We used our computational pipeline developed with Nextflow (https://www.nextflow.io) to integrate gene-expression data with ChIP-Seq and ATAC-Seq data in order to generate a prioritized list of target genes. The change in gene expression (DN3–CD4+) was assessed using publicly available RNA-sequencing data from NCBI-GEO. The processed gene-count table for dataset GSE109125 was imported into R and the raw counts were normalized using the DESeq2 package. Log₂ fold changes in gene expression were computed using the Wald test in DESeq2, and genes with counts less than 10 in all samples were removed from further analysis. The results of the differential-expression analysis were visualized in R with customized volcano plots generated using the ggplot2 graphic package. Processed data from CMS-IP experiments performed in thymocytes were downloaded from Tsagaratou et al.[2]. To identify genomic locations with acetylated histone marks (H3K27), we analyzed publicly available ChIP-Seq data from the NCBI SRA archive. The datasets (SRR5385354, SRR5385355, SRR5385352, and SRR5385353) were downloaded using the SRA toolkit and the SRA files were converted to fastq files for processing. The raw

**Fig. 7 TET-mediated demethylation during thymic development is critical for optimal gene function in effector T cells. a** Venn diagram showing the number of genes that upregulate gene expression (a fold change >2 in gene expression from DN3 to CD4+SP) intersected with genes that have 5hmC in either DP or CD4+ SP T cells (intragenic 5hmC log2 CMS-IP/input >2) intersected with genes containing open-chromatin ATAC-Seq peaks and H3K27Ac marks in either DP or CD4 + T cells. **b** Metascape bar graph showing top nonredundant enrichment clusters among genes that have novel-accessibility ATAC-Seq in activated T cells, undergo DNA demethylation, and upregulate RNA expression during thymic development. The number of genes in each cluster is indicated on the left and cluster IDs are shown on the right. Bar graph was generated using the Metascape gene annotation and analysis online resource. *p*-values were computed using hypergeometric test and Benjamini–Hochberg *p*-value correction algorithm as previously described in the publicly available Metascape interface[72]. List of genes used for the enrichment analysis is found in Supplementary Table 4. **c** FACS plot showing CD4, ThPOK, CD5, and CD6 expression in CFSE-labeled T cells from control or *Rorc(t)$^{CreTg}$ Tet1$^{fl/fl}$ Tet2$^{fl/+}$ Tet3$^{fl/fl}$* mice 72 hrs post activation. Naive CD4 T cells were FACS-sorted and activated with anti-CD3/CD28. Data is a representative of two experiments with at least two animals/genotype/experiment. **d** *Cd4, Zbtb7b, Cd5 and Cd6* mRNA expression in control or *Rorc(t)$^{CreTg}$ Tet1$^{fl/fl}$ Tet2$^{fl/+}$ Tet3$^{fl/fl}$* T cells activated in vitro for 96 hrs with anti-CD3/CD28. Data shown are mean ± SEM (*n* = 3). *p*-values are indicated on graphs (unpaired two-tailed *t*-test). **e** IGV snapshots of the *Zbtb7b* locus depicting p300 and H3K27Ac signals by ChIP-Seq in activated T (Th0) cells and ATAC-Seq peaks in DP, CD4+ thymocytes, and Th0 CD4+ T cells. Red arrows denote currently undefined CREs. Green arrow shows the location of a previously validated CRE PE in CD4+ T cells[61,63]. **f** IGV snapshot of the *Cd5* locus depicting p300 and H3K27Ac signals by ChIP-Seq in activated T (Th0) cells and ATAC Seq peaks in DP, CD4+ thymocytes and Th0 CD4+ T cells. Red arrows denote currently undefined CREs. **g** IGV snapshot of the *Cd6* locus depicting p300 and H3K27Ac signals by ChIP-Seq in activated T (Th0) cells and ATAC-Seq peaks in DP, CD4+ thymocytes and activated CD4+ T cells. Red arrows denote currently undefined CREs.

sequences were first trimmed with fastp and then aligned to the mouse genome (version mm10) using bwa mem. Peak calling was performed with Genrich using the following filtering options: removal of PCR duplicates and retention of unpaired alignments. The default p-value of 0.01 was used for statistical significance. Processed ATAC-Seq data containing peak-location information for immune cells were downloaded from the Immgen project. Peak files from both the ChIP-Seq and ATAC-Seq analyses were imported into R and analyzed in a similar way using the ChIPseeker package. Association of peaks with genes was done using the ChIPseeker annotatePeak function using gene-location data from the UCSC-known gene database. All of the filtering and integration of gene lists from the above three analyses were performed in R. The final list of genes with new accessible ATAC-Seq peaks in Th0 cells was then manually curated to ascertain the presence of peaks within the gene body and <10 kb away from the nearest-annotated gene TSS. Gene-enrichment analysis was performed using the Metascape online platform[72]. p300 and H3K27Ac ChIP-Seq datasets in Th0 cells are from previously published work from Ciofani et al. (GSE40918). Details on the code used to analyze integrated datasets have been deposited at Zenodo (DOI 10.5281/zenodo.5879899)

**Quantification and statistical analysis.** GraphPad Prism v 8.0 or v9.0 software was used for data analysis. Data are represented as the mean ± SEM, unless otherwise specified. Statistical significance was determined by the Mann–Whitney *U*-test for comparison of 2 groups for in vivo analysis or by ANOVA, Student's *t*-test, or multiple-comparison *t*-tests for in vitro experiments as described in the Figure legends. A *p*-value of less than 0.05 was considered statistically significant.

**Reporting summary.** Further information on research design is available in the Nature Research Reporting Summary linked to this article.

## Data availability

Bisulfite datasets using CATCH-Seq have been deposited in the Sequence Read Archive (SRA) under accession code PNJNA765040 (https://www.ncbi.nlm.nih.gov/bioproject/PRJNA765040).

All other datasets used in this paper are publicly available with accession numbers listed above. Source data are provided with this paper. All other relevant data supporting key findings in this study are available in the supplementary information files and further information is available from the corresponding author upon reasonable request. Source data are provided with this paper.

## Code availability

Code used to analyze bioinformatics datasets used in Fig. 7 has been deposited at Zenodo (DOI 10.5281/zenodo.5879899)

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

## Acknowledgements

Some of the data presented herein were obtained at the Flow Cytometry Facility, which is a Carver College of Medicine/Holden Comprehensive Cancer Center core research facility at the University of Iowa. The facility is funded through user fees and the generous financial support of the Carver College of Medicine, Holden Comprehensive Cancer Center, and Iowa City Veteran's Administration Medical Center. We are grateful to Dr. Noah Butler and Jordan Johnson for help with LCMV infections, Dr. Mary Wilson for insights on Leishmaniasis-related experiments, Dr. William Nauseef for insights on the paper, and Dr. Y. Zhang and Dr. I. Aifantis for sharing of TET-mutant mice. We thank Dr. Sangyong Kim and the Rodent Genetic Engineering Core (RGEC) of NYU Medical Center for generation of the *Cd4*-mutant mouse strains. The RGEC is partially supported by Cancer Center Support grant P30CA016087 at the Laura and Isaac Perlmutter Cancer Center of the NYU Medical Center. D.R.L. was supported by the Howard Hughes Medical Institute.

## Author contributions

A.T., P.P., and K.M. designed and performed experiments, analyzed data, and provided input to the paper. C.A. provided technical assistance in the creation and analysis of CRISPR founder mice at NYU. K.D. did methylation captures and bioinformatics analyses of *Cd4* locus-wide methylation captures; H.K. performed the ChIP-Seq alignments, bioinformatics analysis, and integrated datasets for genome-wide analysis. R.Y. provided initial insights on ATAC-Seq analysis. M.Y. performed Leishmaniasis experiments with supervision and insights from PG. P.G. assisted in the interpretation of data and provided input to the paper. TM provided experimental support for multiplex assays and input to the paper. D.R.L. provided guidance and support during the initial conception of this study. P.D.I. designed, performed, and supervised experiments throughout the study, co-analyzed data with A.T., P.P., and K.M., and wrote the paper with input from D.R.L. and co-authors.

## Competing interests

The authors declare no competing interests.
