## [Peer Review File · Nature Communications]

CD4 expression in effector T cells depends on DNA demethylation over a developmentally established stimulus-responsive elementREVIEWER COMMENTS

Reviewer #1 (Remarks to the Author):

In this manuscript Teghanemt and colleagues present a detailed and compelling story involving developmental epigenetic programming of the Cd4 locus that maintains CD4 expression in effector CD4 T cells. The data support a model in which H3K4me3 blocks de novo DNA methylation at critical CREs in the Cd4 locus to maintain CD4 expression. The molecular work is elegant, and demonstrating using cutting-edge in vivo technologies is impactful. I have only a few relatively minor comments:

- 1) The authors state, "our study provides one of the very few demonstrations of the potential for early epigenetically transmitted transcriptional programs that have significant relevance for T cell physiology post-developmentally." While I agree that it is important to conceptually link developmental with maintenance steady state biology, as this study does, there are numerous examples of these phenomena in T cell subsets. I would soften the language here.
- 2) I applaud the authors for testing their mechanisms in an integrated in vivo model, but it is not clear that Leishmaniasis represents the best choice given the relatively unimpressive readouts in Fig 6A-C. I would recommend complementing this in vivo model with some straightforward classic T cell skewing assays in vitro, for example Th2, Th17, and iTreg induction using naive CD4 T cells from the Cd4E4mD/D and E4aD/D mice.
- 3) Figure 7 seems tacked on or at least out of place. I would consider moving this figure to the supplement.
- 4) In the Discussion the authors may want to review the mechanisms demonstrated in Nat Genet. 2020 Jun; 52(6): 615-625.

Reviewer #2 (Remarks to the Author):

This is an elegant study showing that DNA demethylation during thymic development is critical for the licensing of, E4a, a novel stimulus-responsive cis-regulatory element (CRE) that serves to maintain CD4 gene expression in effector T cells. E4a and E4m act in concert to maintain transcriptional activity at the CD4 locus and, thereby, prevent de novo DNA methylation during CD4 T cell proliferation. In absence of thymic demethylation, the authors find that E4a and E4m are less active and CD4 downregulation ensues upon T cell proliferation. Strikingly, this downregulation leads to impaired Th1 differentiation and reduced control of Leishmania infection. This reviewer lacks expertise with respect to the technical validity of some of the assays used by the authors for investigating epigenetic regulation. Still, the experiments appear to be overall very well performed and the conclusions drawn seem valid. In the eyes of this reviewer, the finding that infection control is substantially reduced by joint deletion of E4a and E4m is exciting, as is the association to the Th1 phenotype. And, at least to my knowledge, nothing comparable has been previously shown.

Major Concern:

From an immunological point of view, I think this manuscript would benefit from an additional evaluation of Th2 differentiation and of TCR binding strength to peptide MHCII complexes in E4a/E4m-deficient vs. WT mice.

The Th1/Th2 topic could be tackled via an in vitro differentiation assay using presentation of actual peptide-MHCII complexes for stimulation (and not CD3) or by using a Th2 infection model in addition to the Th1-controlled Leishmania model or by asking, whether during Leishmania infection there is suddenly aberrant Th2-cytokine production.

The binding strength topic could be investigated (at least roughly) by measuring functional avidity through peptide titration upon T cell restimulation in the Leishmania system and then calculating an EC50 for IFN γ secretion, or in absence of this cytokine of CD107 degranulation of CD69 upregulation.

The concept to be tested would be: Does CD4 downregulation reduce TCR binding strength to peptide MHCII complexes and, thereby, favor Th2 vs. Th1 differentiation. See e.g. (Van Panhuys et al., 2014).

Van Panhuys, N., Klauschen, F., and Germain, R.N. (2014). T-cell-receptor-dependent signal intensity dominantly controls CD4(+) T cell polarization In Vivo. *Immunity* 41, 63–74.

Reviewer #3 (Remarks to the Author):

It is not uncommon to hear the term epigenetic regulation of gene expression used to describe chromatin effects on gene activity. Yet, epigenetic effects typically imply an inducible change in the activation or repression of a gene that can be heritably and stably transmitted across the space and time of cell division and cell migration; often transcending the inductive signal itself. How such signaling behaviors are remembered through development, especially in mobile cells, has remained obscure. Teghanemt et al. demonstrate that a cis-acting regulatory element of the Cd4 locus undergoes programmed demethylation during thymic development of CD4+ T cells. The primary function of this inductive event in the thymus is to pre-configure a permissive chromatin landscape such that the cellular descendants of a naive CD4 T cell maintain gene activity and function after the naive cell's bona fide antigen activation and the subsequent dispatch of those daughter cells to peripheral tissues. The novel cis-acting regulatory element (E4a) works in concert with

with a developmentally important cis-acting regulatory element (E4m) to heritably maintain CD4 expression in the descendant of the antigen-activated naive T cell. Mechanistically, the authors demonstrate that TET1/3 induces the DNA demethylation in the thymus and without this developmental activation, peripheral CD4 T cells undergo de novo methylation and silencing of the Cd4 locus during their post-antigen-activation cell divisions. The data quality is excellent and the conclusions are supported by the results. This study will be an important new paradigm in understanding epigenetic gene regulation.

Reviewer #1 (Remarks to the Author):

In this manuscript Teghanemt and colleagues present a detailed and compelling story involving developmental epigenetic programming of the Cd4 locus that maintains CD4 expression in effector CD4 T cells. The data support a model in which H3K4me3 blocks de novo DNA methylation at critical CREs in the Cd4 locus to maintain CD4 expression. The molecular work is elegant, and demonstrating using cutting-edge in vivo technologies is impactful. I have only a few relatively minor comments:

Response:

We thank the reviewer for the positive feedback on our work.

1) The authors state, “our study provides one of the very few demonstrations of the potential for early epigenetically transmitted transcriptional programs that have significant relevance for T cell physiology post-developmentally.” While I agree that it is important to conceptually link developmental with maintenance steady state biology, as this study does, there are numerous examples of these phenomena in T cell subsets. I would soften the language here.

Response:

In response to the reviewer’s comment, we have edited the aforementioned statement to read as “our study provides a mechanistic demonstration of the potential for early epigenetically transmitted transcriptional programs that have significant relevance for T cell physiology post-developmentally”.

2) I applaud the authors for testing their mechanisms in an integrated in vivo model, but it is not clear that Leishmaniasis represents the best choice given the relatively unimpressive readouts in Fig 6A-C. I would recommend complementing this in vivo model with some straightforward classic T cell skewing assays in vitro, for example Th2, Th17, and iTreg induction using naïve CD4 T cells from the Cd4E4mD/D and E4aD/D mice.

Response:

We are thankful for the reviewer’s suggestion. Given that *in vitro* T cell skewing assays require the use of anti-CD3/anti-CD28-mediated TCR activation and bypass CD4 co-engagement, we did not anticipate any differences between WT and E4a nor E4a/E4m-deficient T cells when driven to differentiate into T helper subsets. Nonetheless, in the event of an unexpected contribution for CD4 in that system, we have performed such assays but did not find any differences in Th1, Th2, Treg nor Th17 differentiation, as assessed by master transcription factor expression of the respective lineages. To keep our study focused, we have only included Th1 and Th2 differentiation results in the manuscript (**Supplementary Fig. 6j, k**). Moreover, in addition to the Leishmaniasis

system, we have now included a second model of Th1 differentiation using an acute viral pathogen (LCMV Armstrong), which triggers a robust Th1 effector response (**Supplementary Fig. 6f**), and uncovered a similar defect in Th1 differentiation (**Supplementary Fig. 6g**), as observed in the Leishmaniasis system. Together, we hope the reviewer's concerns are adequately addressed.

3) Figure 7 seems tacked on or at least out of place. I would consider moving this figure to the supplement.

Response:

We believe that Figure 7, together with Supplementary Figure 7, extend the idea that many genes in CD4⁺ T cells, beyond the dissected example of the *Cd4* gene, undergo similar patterns of regulation during thymic development and is relevant to the general audience. We strongly believe that it adds significant value to our working paradigm and moving this entire figure to a supplementary will be a pity as it may be overlooked by the readership.

4) In the Discussion the authors may want to review the mechanisms demonstrated in Nat Genet. 2020 Jun;52(6):615-625.

Response:

We sincerely thank the reviewer for bringing our attention to this elegant work and agree that the mechanisms proposed are highly relevant and compatible with our working model that H3K4me3 does not dictate coding gene transcription per say but has an instructional role in preventing repression of gene expression via DNA methylation. We have now included this reference to our discussion in the manuscript.

Reviewer #2 (Remarks to the Author):

This is an elegant study showing that DNA demethylation during thymic development is critical for the licensing of, E4a, a novel stimulus-responsive cis-regulatory element (CRE) that serves to maintain CD4 gene expression in effector T cells. E4a and E4m act in concert to maintain transcriptional activity at the CD4 locus and, thereby, prevent de novo DNA methylation during CD4 T cell proliferation. In absence of thymic demethylation, the authors find that E4a and E4m are less active and CD4 downregulation ensues upon T cell proliferation. Strikingly, this downregulation leads to impaired Th1 differentiation and reduced control of *Leishmania* infection.

This reviewer lacks expertise with respect to the technical validity of some of the assays used by the authors for investigating epigenetic regulation. Still, the experiments appear to be overall very well performed and the conclusions drawn seem valid. In the eyes of this reviewer, the finding that infection control is

substantially reduced by joint deletion of E4a and E4m is exciting, as is the association to the Th1 phenotype. And, at least to my knowledge, nothing comparable has been previously shown.

Response:

We thank the reviewer for the positive feedback on our work.

Major Concern:

From an immunological point of view, I think this manuscript would benefit from an additional evaluation of Th2 differentiation and of TCR binding strength to peptide MHCII complexes in E4a/E4m-deficient vs. WT mice.

The Th1/Th2 topic could be tackled via an *in vitro* differentiation assay using presentation of actual peptide-MHCII complexes for stimulation (and not CD3) or by using a Th2 infection model in addition to the Th1-controlled Leishmania model or by asking, whether during Leishmania infection there is suddenly aberrant Th2-cytokine production.

The binding strength topic could be investigated (at least roughly) by measuring functional avidity through peptide titration upon T cell restimulation in the Leishmania system and then calculating an EC50 for IFN γ secretion, or in absence of this cytokine of CD107 degranulation or of CD69 upregulation. The concept to be tested would be: Does CD4 downregulation reduce TCR binding strength to peptide MHCII complexes and, thereby, favor Th2 vs. Th1 differentiation. See e.g. (Van Panhuys et al., 2014).

Van Panhuys, N., Klauschen, F., and Germain, R.N. (2014). T-cell-receptor-dependent signal intensity dominantly controls CD4(+) T cell polarization *In Vivo*. *Immunity* 41, 63–74.

Response:

We thank the reviewer for the questions posed and recommendations to tackle them. We wanted to evaluate the consequences of reduced CD4 expression on TCR binding and downstream signaling, to support the presented evidence that TCR signaling strength is decreased in antigen-experienced E4a/E4m-deficient T cells during Leishmaniasis, by virtue of CD5 and Nur77 expression levels. However, in the absence of a transgenic system (e.g. an OTII-TCR Tg system), it is challenging to test the question of functional avidity directly. Using such an *in vitro* TCR Tg system is best suited for robust conclusions because we would be able to a) bypass any differences in helper T cell differentiation that happens between WT and E4a/E4m-deficient mice during infection b) then specifically assess the contribution of CD4 expression on functional avidity through peptide titration and assess an EC50 value for cytokine responses such as IL-2. We were not successful in establishing a reliable dose-response using crude soluble leishmania antigen (SLA) extracts in a WT setting, as the

response was too variable, highlighting the need for defined antigen epitopes and specific TCR strategies.

Nevertheless, we have addressed some aspects of the questions raised using a different approach. We have challenged E4a/E4m-deficient and WT mice with an LCMV Armstrong strain, which induces an acute Th1 CD4 response in addition to a cytotoxic CD8 response. We chose to do so because of the well-characterized expansion of a dominant MHC-Class II-restricted (IA^b- restricted) CD4 T cell clone (SMARTA-TCR) that recognizes a specific LCMV glycoprotein epitope GP 61-80. Using tetramer tools (GP 66-77) and a peptide specific to this TCR clone to then engage stimulation, we were then able to ask a) does reduced CD4 expression result in a compromised ability to bind to antigen b) is there an impact on Th1 differentiation c) is there a diversion to a Th2 response. To summarize, we found that while a reduction in CD4 expression did not significantly impair the ability of CD4 T cells to bind to GP66-77 tetramers, a complete loss of CD4 expression had a modest but significant impact (**Fig. 6i, j, k**). We found a significant reduction in Tbet⁺ CD4⁺ T cells in LCMV-infected E4a/E4m-deficient mice but no significant increase in Gata3⁺CD4⁺ T cells (**Supplementary Fig. 6g, h**). Accordingly, E4m/E4a-deficient antigen-experienced CD4⁺ T cells mounted a less robust IFN- γ response upon antigen re-stimulation (**Fig. 6i**) but we detected no significant IL-4 production (data not shown). Together, these findings indicate that reduced TCR signaling mediated via CD4 impairs Th1 differentiation but does not result in preferential Th2 differentiation and are in agreement with our findings in the Leishmaniasis model, where we did not observe an increase in Gata3⁺ T cells nor an increase in IL-4 production in the footpads (**Supplementary Fig. 6b, d, e**).

These results may highlight important differences in modulating TCR signaling via alterations in CD4 levels (which in our model is mostly affected during T cell replication) versus pMHC density as demonstrated by Van Panhuys et al., 2014 and/or other environmental/ cellular factors. We agree that it would be exciting to test whether CD4 expression has an impact on Th2 differentiation in the context of a Th2-dominant infection model. However, we hope the reviewer agrees this merit a future study.

Reviewer #3 (Remarks to the Author):

It is not uncommon to hear the term epigenetic regulation of gene expression used to describe chromatin effects on gene activity. Yet, epigenetic effects typically imply an inducible change in the activation or repression of a gene that can be heritably and stably transmitted across the space and time of cell division and cell migration; often transcending the inductive signal itself. How such signaling behaviors are remembered through development, especially in mobile cells, has remained obscure. Teghanemt et al. demonstrate that a cis-acting regulatory element of the Cd4 locus undergoes programmed demethylation during thymic development of CD4⁺ T cells. The primary function of this inductive event in the thymus is to pre-configure a permissive chromatin landscape such

that the cellular descendants of a naive CD4 T cell maintain gene activity and function after the naive cell's bona fide antigen activation and the subsequent dispatch of those daughter cells to peripheral tissues. The novel cis-acting regulatory element (E4a) works in concert with a developmentally important cis-acting regulatory element (E4m) to heritably maintain CD4 expression in the descendant of the antigen-activated naive T cell. Mechanistically, the authors demonstrate that TET1/3 induces the DNA demethylation in the thymus and without this developmental activation, peripheral CD4 T cells undergo de novo methylation and silencing of the Cd4 locus during their post-antigen-activation cell divisions. The data quality is excellent and the conclusions are supported by the results. This study will be an important new paradigm in understanding epigenetic gene regulation.

Response:

We are sincerely thankful to the reviewer for the positive and insightful remarks on our work.

REVIEWERS' COMMENTS

Reviewer #1 (Remarks to the Author):

The authors have satisfactorily addressed my comments. I congratulate the team on a beautiful paper.

Reviewer #2 (Remarks to the Author):

The authors have sufficiently addressed my concerns. I do not have any further comments.

Reviewer #3 (Remarks to the Author):

All 3 reviewers acknowledged the elegance and importance of this study and the revised manuscript has been highly responsive to the critiques. This will be an important contribution to the field.

Reviewer #1 (Remarks to the Author):

The authors have satisfactorily addressed my comments. I congratulate the team on a beautiful paper.

Response:

We thank the reviewer for their time and positive remark on our work.

Reviewer #2 (Remarks to the Author):

The authors have sufficiently addressed my concerns. I do not have any further comments.

Response:

We are thankful to the reviewer for their time and pleased that we were able to address all their concerns.

Reviewer #3 (Remarks to the Author):

All 3 reviewers acknowledged the elegance and importance of this study and the revised manuscript has been highly responsive to the critiques. This will be an important contribution to the field.

Response:

We are thankful to the reviewer for their time and are greatly invigorated by their positive outlook on our study.